# Enhancing Long-Term Vegetation Monitoring in Australia: A New Approach for Harmonising and Gap-Filling AVHRR and MODIS NDVI

5  Chad. A. Burton[1], Sami. W. Rifai[2], Luigi. J. Renzullo[3] and Albert. I.J.M. Van Dijk[1]

[1] Fenner School of Environment & Society, Australian National University, Canberra, ACT, Australia.

[2] School of Biological Sciences, The University of Adelaide, Adelaide SA, Australia.

[3] Bureau of Meteorology, Hydrology Science, Canberra, Australia

*Correspondence*: Chad Burton (chad.burton@anu.edu.au)

**Abstract.**

Long-term, reliable datasets of satellite-based vegetation condition are essential for understanding
terrestrial ecosystem responses to global environmental change, particularly in Australia which is characterised by diverse ecosystems and strong interannual climate variability. We comprehensively evaluate several existing global AVHRR NDVI products for their suitability for long-term vegetation monitoring in Australia. Comparisons with MODIS NDVI highlight significant deficiencies, particularly over densely vegetated regions. Moreover, all the assessed products failed to adequately reproduce inter-
annual variability in the pre-MODIS era as indicated by Landsat NDVI anomalies. To address these limitations, we propose a new approach to calibrating and harmonising NOAA's Climate Data Record AVHRR NDVI to MODIS MCD43A4 NDVI for Australia using a gradient-boosting decision tree ensemble method. Two versions of the datasets are developed, one incorporating climate data in the predictors ('AusENDVI-clim': **Aus**tralian **E**mpirical **NDVI-clim**ate) and another independent of climate
data ('AusENDVI-noclim'). These datasets, spanning 1982-2013 at a spatial resolution of 0.05° and monthly time step, exhibit strong correlation ($r^2$ = 0.89-0.94) and low mean errors compared to MODIS MCD43A4 NDVI (MAE = 0.014-0.028, RMSE = 0.021-0.046), accurately reproducing seasonal cycles over densely vegetated regions. Furthermore, they closely replicate the interannual variability in

vegetation condition in the pre-MODIS era. A reliable method for gap-filling the AusENDVI record is also developed that leverages climate, atmospheric $CO_2$ concentration, and woody cover fraction predictors. The resulting synthetic NDVI dataset shows excellent agreement with observations ($r^2 = 0.82$-$0.95$, MAE = 0.016-0.029, RMSE = 0.039-0.041). Finally, we provide a complete 41-year dataset where gap filled AusENDVI from January 1982 to February 2000 is seamlessly joined with MODIS MCD43A4 NDVI from March 2000 to December 2022. Analysing 40-year per-pixel trends in Australia's annual maximum NDVI revealed increasing values across most of the continent. Shifts in the timing of annual peak NDVI are also identified, underscoring the dataset's potential to address crucial questions regarding changing vegetation phenology and its drivers. The AusENDVI dataset can be used for studying Australia's changing vegetation dynamics and downstream impacts on terrestrial carbon and water cycles, and provides a reliable foundation for further research into the drivers of vegetation change. AusENDVI is open access and available at https://doi.org/10.5281/zenodo.10802704 (Burton, 2024)

## 1 Introduction

Australia is undergoing long-term changes to its climate that are impacting terrestrial vegetation, with attendant serious implications for ecosystem functioning, carbon and water cycles, and agriculture (Hoffmann et al., 2019; Canadell et al., 2021; Head et al., 2014; Hughes, 2011; Steffen et al., 2011; Rifai et al., 2022; Ukkola et al., 2016; Donohue et al., 2009). Long-term, reliable datasets that chart the land surface response to climate change are crucial if we are to identify, understand, and respond to ongoing terrestrial ecosystem change (Giglio and Roy, 2020; Piao et al., 2019). One of the primary means Earth System Science has to trace long-term vegetation change is the Normalised Difference Vegetation Index (NDVI), a widely used satellite-derived indicator of vegetation condition owing to its close relation to vegetation productivity. NDVI provides an efficient means for mapping and monitoring vegetation condition at continental scales. In Australia, the need for very long records of NDVI to understand change is amplified by strong variability at both interannual and interdecadal time scales, and ecosystems that are often driven by periodic, but non-seasonal phenological drivers (Moore et al., 2016; Chambers et al., 2013; Ma et al., 2013; Beringer et al., 2022).

The MODerate resolution Imaging Spectroradiometer (MODIS) NDVI record ($NDVI_{MODIS}$) is generally considered the most reliable global scale dataset due to its high quality radiometrics and accurate georeferencing. Unfortunately, the MODIS record only begins in March 2000 (Vermote et al., 2002). The Advanced Very High-Resolution Radiometer (AVHRR) NDVI record ($NDVI_{AVHRR}$) is the longest contiguous series of satellite data, starting in July 1981, but has several well-known problems owing to a lack of on-board calibration for visible wavelengths, sensor orbital drift, and sensor degradation, making it unreliable for detecting relatively subtle trends over multiple decades (Tucker et al., 2005; Privette et al., 1995; Gorman and Mcgregor, 1994). Several prominent global $NDVI_{AVHRR}$ products attempt to ameliorate these issues. For example, the Global Inventory Modelling and Mapping Studies version 3 ($NDVI_{GIMMS3g}$) applies Bayesian analysis with Sea-Viewing Wide Field-of-View Sensor NDVI as evidence information to reduce sensor transition discontinuities and increase the dynamic range of $NDVI_{AVHRR}$ (Pinzon and Tucker, 2014), while the NOAA Climate Data Record ($NDVI_{CDR}$) applies a suite of corrections to create a consistent surface reflectance product (Franch et al., 2017), among others (Table 1). However, despite substantial progress, errors and biases in these NDVI products have led to inconsistent findings on global greening (Wang et al., 2022; Wang et al., 2021; Cortés et al., 2021; Frankenberg et al., 2021; Fensholt and Proud, 2012), discrepancies in vegetation seasonality between datasets (Ye et al., 2021), and persistent temporal inconsistencies (Tian et al., 2015; Giglio and Roy, 2020). Recently, Li et al. (2023) developed a new global $NDVI_{AVHRR}$ product, 'GIMMS-PKU' ($NDVI_{GIMMS-PKU}$), which effectively calibrates the $NDVI_{GIMMS3g}$ archive to the Landsat record using machine learning techniques, and 'GIMMS-PKU-consolidated' ($NDVI_{PKU-consolidated}$) which harmonises $NDVI_{GIMMS-PKU}$ to $NDVI_{MODIS}$ (Table 1), but which has yet to be extensively assessed in the literature (Li et al., 2023).

As much as possible, any NDVI product that exploits the AVHRR and MODIS record to acquire an accurate >40-year record of vegetation condition should attempt to integrate the two seamlessly while also performing well in the pre-MODIS AVHRR era (1982-2000). Performance should be judged on how well seasonal cycles are represented along with interannual and interdecadal variability, as both seasonal and longer-term fluctuations in vegetation conditions have important ramifications for carbon and water cycles (Ma et al., 2015). An effectively calibrated, harmonised, and gap-filled dataset can form

the basis for studying the biogeophysical impacts of global change and meteorological variability on Australia's terrestrial vegetation. With that in mind, the objectives of this study are as follows:

- To investigate existing $NDVI_{AVHRR}$ datasets to determine their suitability for long-term vegetation monitoring in Australia by both comparing their consistency with $NDVI_{MODIS}$ during the 2000-2013 overlap period, and with Landsat NDVI ($NDVI_{Landsat}$) anomalies from 1988-2000.

- Having established limitations with the existing datasets, calibrate and harmonise $NDVI_{AVHRR}$ to $NDVI_{MODIS}$ solely over Australia at the highest spatial resolution possible. The final dataset should contain the harmonised $NDVI_{AVHRR}$ from January 1982 to February 2000, where it seamlessly joins with the superior $NDVI_{MODIS}$ timeseries, resulting in a reliable 41-year record of vegetation condition for Australia. We will call this time series "AusENDVI" (for Australian Empirical NDVI; $NDVI_{AusE}$)

- To develop a reliable method for gap filling the $NDVI_{AusE}$ record caused by sensor transitions issues and long periods of missing or suspect data acquisition.

- To demonstrate the utility of this new dataset by exploring NDVI phenology trend analysis, including long-term trends in the value and timing of annual maximum NDVI across the Australian continent.

## 2 Materials and Methods

### 2.1 Datasets

Specifications of all datasets used for either the intercomparison of NDVI products or in the modelling framework are listed in Table 1. For comparisons between NDVI datasets, finer resolution datasets were resampled to match the coarsest grid (i.e., GIMMS, 1/12º or ~8 km over Australia). Averaging resampling techniques were used for downsampling finer-resolution datasets, while nearest-neighbour techniques were used when datasets have a similar spatial resolution but either different projections or slightly different grid extents. Wherever datasets are compared, data gaps are matched between all datasets by creating a mask that identifies all missing pixels, and then that common mask is applied to every dataset. This ensures a fair and valid comparison. We chose Landsat's TM and ETM+ (Table 1) as the sensor for

comparison in the pre-MODIS era owing to the international efforts to produce a relatively high geometric and radiometric accuracy for its generation, and its lack of sensor transitions in the pre-MODIS era from 1982-1999 (Beck et al., 2011). The chosen surface reflectance Landsat product, Digital Earth Australia's (DEA) Landsat NBAR (Nadir-corrected BRDF Adjusted Reflectance, where BRDF stands for Bidirectional reflectance distribution function) product is calibrated to Australia's environment using the MODTRAN 4 radiative transfer model and BRDF shape functions derived from MODIS (Li et al., 2010; Byrne et al., 2024).

For the development of the Australian NDVI dataset, we relied on the NOAA $NDVI_{CDR}$ product (Franch et al., 2017) as the input dataset. This was principally because of its higher spatial resolution than the other datasets (~5 km), its lack of gap filling, extensive atmospheric corrections, and its BRDF-based correction of view-angle effects (Ma et al., 2019). As the target dataset, we derived NDVI from the MODIS MCD43A4 surface reflectance NBAR product ($NDVI_{MCD43A4}$). This reflectance product was chosen because of its similar set of atmospheric corrections when compared with $NDVI_{CDR}$ and DEA's Landsat NBAR, and its use of both the Terra and Aqua instruments which extends its temporal extent back to March 2000 (Schaaf and Wang, 2015).

All additional input data used in NDVI estimation were temporally aggregated to monthly values by calculating medians and spatially reprojected onto a common $0.05°$ geographic grid. In addition to filtering based on the quality assurance band (we filtered for clouds, cloud shadows, and invalid BRDF and channel values) additional criteria were applied to minimise the impact of temporal discontinuities in the $NDVI_{CDR}$ record that may arise from orbital decay or sensor degradation. Monthly $NDVI_{CDR}$ values based on fewer than two observations per month were discarded, along with any values for which the coefficient of variation in daily retrievals for a given month was greater than 50 %. Anomalies in NDVI, solar-zenith-angle, and time-of-acquisition that were greater than 3.5 standard deviations were also discarded (based on a 1982-2013 climatology). Following the advice of Tian et al. (2015), data for several problematic sensor transition periods were discarded (September 1984 - April 1985, July 1988 - September 1989, and July 1993 - December 1994). After filtering, the continental average fraction of available data is 0.79, meaning on average 79 % of the monthly time-steps between 1982-2013 are preserved (Fig. A1).

**Table 1: Details of the datasets used in, and produced by, this study.**

| Dataset & Abbreviation | Native spatial resolution; temporal resolution & range; additional details | Data Source & Reference |
|---|---|---|
| AVHRR Climate Data Record NDVI and Surface Reflectance; NDVI$_{CDR}$ | 0.05º, Daily, January 1982 to December 2013. Surface reflectance product used for the time-of-day and solar zenith angle. | Version 5, downloaded from Google Earth Engine, (Franch et al., 2017) |
| MODIS MCD43A4 NDVI; NDVI$_{MCD43A4}$ | ~500m, 16-day, March 2000 to December 2022. Calculated from the combined Terra and Aqua MCD43A4 surface reflectance NBAR product. | Version 6.1 downloaded from Google Earth Engine (Schaaf and Wang, 2015) |
| AVHRR GIMMS3g NDVI; NDVI$_{GIMMS3g}$ | 1/12º, Half-month, 1982-2013. AVHRR NDVI with sensor transition discontinuities reduced with Bayesian analysis. | Version 1.0 downloaded from Google Earth Engine (Pinzon and Tucker, 2014) |
| AVHRR GIMMS PKU NDVI; NDVI$_{PKU}$, NDVI$_{PKU\text{-}consolidated}$ | 1/12º, Half-month, 1982-2022. Two variations, 'GIMMS-PKU-solely' and 'GIMMS-PKU-consolidated', the latter is harmonised with MODIS MOD13C1. For GIMMS-PKU-solely we loaded pixels labelled as 'good-quality AVHRR'. For GIMMS-PKU-consolidated we loaded pixels labelled as 'good-quality AVHRR' and 'good-quality MODIS' and where the harmonisation was run by the random-forest model | Version 1.2 downloaded from https://zenodo.org/records/8253971 (Li et al., 2023) |
| Digital Earth Australia's Landsat NDVI (NBAR); NDVI$_{Landsat}$ | 30m, 8-day, 1987-2012, NDVI calculated from an Australian-specific Landsat 5 & 7 surface reflectance NBAR product. | Collection 3, https://docs.dea.ga.gov.au/data/product/dea-surface-reflectance-nbar-landsat-5-tm/ (Li et al., 2010) |
| AusENDVI-clim and AusENDVI-noclim; NDVI$_{AusE\text{-}clim}$, NDVI$_{AusE\text{-}noclim}$ | 0.05º, Monthly, 1982-2013. Calibrated and harmonised NDVI for Australia using machine-learning techniques. The 'clim' version of the dataset includes climate variables in the feature set, the 'noclim' version does not. | This study |
| Synthetic NDVI; NDVI$_{SYN}$ | 0.05º, Monthly, 1982-2022. A machine-learning derived synthetic NDVI built using climate, $CO_2$, and landscape features, and trained on NDVI$_{AusE\text{-}clim}$ and NDVI$_{MCD43A4}$. | This study |

| ANU Climate:<br>• Average Air Temp; Tavg<br>• Vapour Pressure Deficit; VPD<br>• Incoming Shortwave Radiation; srad<br>• Total Precipitation; rain | ~1 km, Monthly, 1982-2022. Gridded climate products based on topographically conditional spatial interpolation of weather stations. | ANUClimate,<br>https://dapds00.nci.org.au/thredds/catalogs/gh70/catalog.html<br>(Hutchison et al., 2014) |
| --- | --- | --- |
| Atmospheric $CO_2$ concentration | N/A., Monthly, 1982-2022. Extracted from the Cape Grim Baseline Air Pollution Station in Tasmania, Australia. De-seasonalised using a 12-month running mean. | CSIRO Environment and the Australian Bureau of Meteorology (Kennaook / Cape Grim Baseline Air Pollution Station).<br>https://capegrim.csiro.au/ |
| Woody Cover Fraction; WCF | 25m, Annual, 1982-2022. A per-pixel estimate of woody cover fraction across Australia. Annual product for 1990-2022. A five-year average from 1990-1995 was used to extend the product back to 1982. | https://dapds00.nci.org.au/thredds/catalog/ub8/au/LandCover/DEA_ALC/catalog.html<br>(Liao et al., 2020) |

## 2.2 Assessment of existing NDVI products

We compared $NDVI_{AVHRR}$ datasets with $NDVI_{MCD43A4}$ for the overlapping period from March 2000 to December 2013. Per-pixel Pearson correlation (r) and coefficient of variation (CV; root mean square error divided by the long-term mean $NDVI_{MCD43A4}$) describe the agreement between datasets, in addition

to comparison of the long-term seasonal cycle. Next, $NDVI_{AVHRR}$ datasets were compared to rolling annual mean 'z-score' standardised anomalies of $NDVI_{Landsat}$ for 1988-2000 to assess how well each product reproduces inter-annual variability in vegetation condition in the pre-MODIS era. Z-score standardised anomalies are calculated as $(x - \mu) / \sigma$ where x is a monthly NDVI observation, $\mu$ is the long-term mean NDVI for the given month, and $\sigma$ is the long-term standard deviation in NDVI for the given

150 month. Differences in spectral sampling between MODIS and Landsat result in differences in mean NDVI so we use Landsat only for validating inter-annual variability in the pre-MODIS era since mean differences in NDVI between sensors are removed by anomalies. We compared NDVI anomalies in $NDVI_{Landsat}$ with $NDVI_{MCD43A4}$ during an overlap period from 2000-2012 to ensure $NDVI_{Landsa}$ could

provide a consistent evaluation of interannual variability in the pre-MODIS era and found good agreement
between the two products (Fig. A2).

## 2.3 Calibration and harmonisation

During extensive preliminary testing gradient-boosting decision tree ensembles (GBM), random forest, and generalised additive models were assessed for their ability to calibrate and harmonise $NDVI_{CDR}$ with $NDVI_{MCD43A4}$. The GBM outperformed the other approaches. Two classes of models and datasets were built: one that utilises climate data (hereafter referred to as 'clim' models) in the feature set to achieve the best possible agreement between $NDVI_{CDR}$ and $NDVI_{MCD43A4}$. The second excludes climate features (hereafter, 'noclim' model) while still achieving satisfactory results. When examining drivers of change, users of these datasets may prefer to use the no-climate model to limit potential circularities in attribution of the drivers of change. During testing, climate variables were identified as useful features for both improving predictions in the heavily forested regions where there was little to no agreement between $NDVI_{MCD43A4}$ and $NDVI_{AVHRR}$, and for capturing interannual variability. The 'noclim' models used the following features: solar-zenith-angle (SZEN), time-of-acquisition (TOD), month-of-year, latitude, and $NDVI_{MCD43A4}$ summary percentiles (0.05, 0.5, and 0.95). $NDVI_{MCD43A4}$ summary percentiles were calculated per pixel over the 2000-2022 period. The 'clim' models used the same variables, plus incoming solar radiation, rainfall, temperature, and vapour pressure deficit. Fractional anomalies of the climate features are also included, along with cumulative three- and six-month rainfall. Testing revealed the best results were obtained by generating three separate models for areas with high and low woody cover fraction (WCF), and for the desert bioclimatic region (Fig. 1a and Fig. 2a). The long-term mean of WCF was extracted from Liao et al. (2020) and a threshold of WCF=0.25 was used to separate regions with a high woody canopy cover. This threshold was chosen as it approximately delineated those regions with the poorest correspondence between $NDVI_{CDR}$ and $NDVI_{MCD43A4}$ (Figure 3e-h).

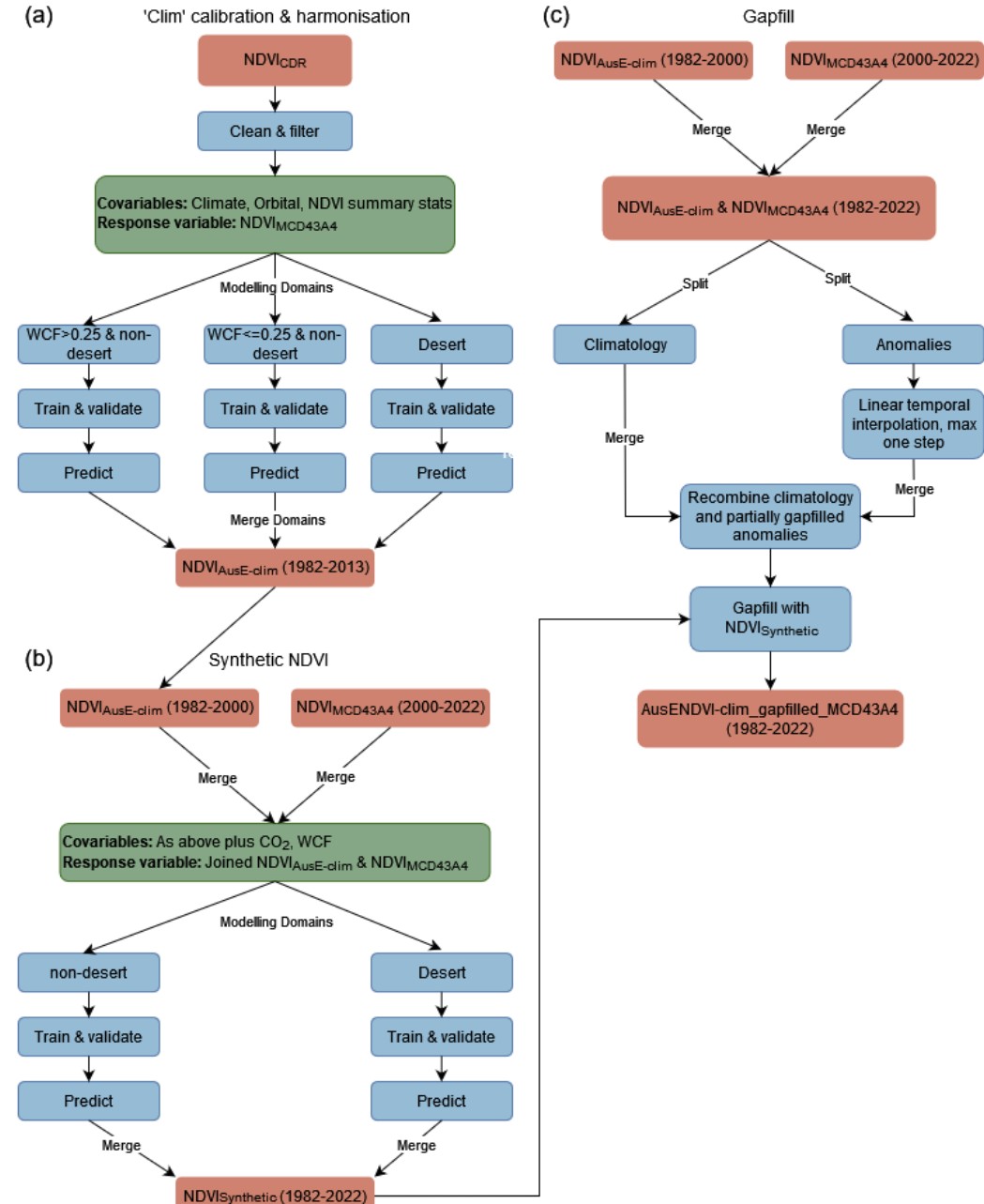

**Figure 1: Flowchart describing the calibration and harmonisation methods (a), and the development of a synthetic NDVI (b) for gap filling (c). a) Shows the method for the 'clim' model type, the methods for 'noclim' are the same but climate variables are removed from the covariables and 'noclim' is not gap filled. Red coloured boxes denote datasets, blue boxes denote processing steps, and green boxes describe the response variables and covariables used for modelling.**

Owing to the differing volumes of good quality data across the continent (Fig. A1) and the large

difference in land area of each bioclimatic region, we implemented a stratified, equalised random sampling approach for the training and validation samples to reduce bias in the sample allocations. In the high and low WCF regions, 30,000 training and testing samples were extracted in equal measure from the five remaining bioclimatic regions after excluding the desert (i.e., 6,000 samples per region). Bioclimatic regions were identical to those defined by Haverd et al. (2013) (Fig. 2b). In the desert region, samples

were drawn using a simple random approach. In all modelling domains, samples were drawn from any point in time across the overlap period, and 5,000 samples were randomly separated as an independent validation set, leaving 25,000 samples for training. The calibration and harmonisation process are summarised in the flow chart of Figure 1a.

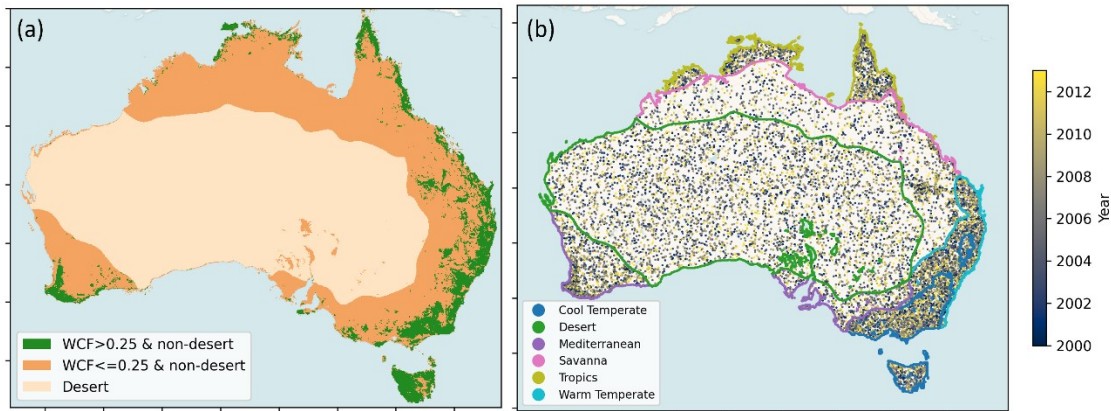

**Figure 2: a) Regions delineating the spatial extent of the three modelling domains: desert, low woody cover fraction (WCF) and high WCF. b) The distribution of all independent validation points used to assess the model fits across the three modelling domains in (a); points are coloured by the year they are drawn from. Figure is overlaid with outlines of the six bioclimatic regions used to both stratify training points and for aggregating trends in later analysis.**

Cross validation for model hyperparameter optimization was conducted using a nested cross-validation approach with five outer splits and three inner splits (Cawley and Talbot, 2010), the hyperparameter grid search parameters are listed in Table A1. Mean absolute error (MAE), root mean square error (RMSE), and the coefficient of determination ($r^2$) are reported as indicators of the goodness

of fits. To understand which explanatory variables most impacted predictions, feature importance plots

were produced using the Shapley Additive Explanations (SHAP) Python library (Lundberg and Lee, 2017).

## 2.3 Gap-filling

At times there are long gaps in AVHRR data acquisition over Australia. For example, 1994 is entirely missing, and during sensor transition periods the data becomes unreliable for several months before and after the transitions (Tian et al., 2015). Furthermore, owing to the nature of Australia's prevailing weather systems such as the tropical monsoon, it is not uncommon to have whole geographic regions missing for a given month. This undermines the typical approaches to gap filling that work well when either the temporal gap is short (e.g., temporal interpolation methods using linear or polynomial fits), or the spatial pattern of gaps are quasi-random such as from scattered cloud cover (spatial interpolation methods such as nearest neighbour, kriging etc.) (Bessenbacher et al., 2022; Shen et al., 2015). Gap-filling with a climatology can often mask important interannual variability at key times – such as anomalously high rainfall periods associated with La Niñas when enhanced cloud cover masks large-scale greening events across Australia's northern tropical savanna. To avoid this we used well established machine learning approaches that have been developed to fill gaps in univariate data (Gerber et al., 2018; Zeng et al., 2014). Here, we develop a two-stage process for gap-filling (summarised in Fig. 1b-c). Firstly, to fill short temporal gaps, the time series is split into a climatology and anomaly series and linear temporal interpolation is applied to the anomalies for a maximum of one time-step (i.e., one month). Longer temporal gaps are replaced with a synthetic NDVI dataset generated using a similar GBM machine learning method as the harmonisation and is described further below.

### 2.3.1 Synthetic NDVI

Training samples were extracted from $NDVI_{AusE\text{-}clim}$ for 1982-2000 and $NDVI_{MCD43A4}$ for 2000-2022, using a similar sampling approach as used for harmonisation only in this instance two models are built, a 'desert' model and 'non-desert' model. The non-desert model covers the same region as the high and low WCF models previously described (the inclusion of WCF in the features reduces the need to define a low and high WCF modelling region). GBM models were then fit using all the features previously listed for

the 'clim' model, plus de-seasonalised $CO_2$ concentration and annual WCF. Otherwise, the modelling framework was the same as the harmonisation approach (Fig. 1b). The synthetic NDVI datasets ($NDVI_{SYN}$) are used to gap fill the $NDVI_{AusE-clim}$ record from January 1982 to February 2000. The final gap-filled, calibrated, and harmonised $NDVI_{AusE-clim}$ dataset is joined with $NDVI_{MCD43A4}$. Only the $NDVI_{AusE-clim}$ dataset is gap filled, the $NDVI_{AusE-noclim}$ dataset is simply joined with the $NDVI_{MCD43A4}$ record. This ensures the 'noclim' dataset does not contain any climate information in the reconstructed time series.

## 2.4 Trends in peak-of-season phenology

Annual, per-pixel NDVI land surface phenology statistics were extracted using the "xr_phenology" Python function from the "dea-tools" package (Krause et al., 2021). This analysis focused on two metrics, the NDVI value at the peak of the season (vPOS), and the day-of-year the peak occurs (POS). The input time-series was the gap-filled 'clim' dataset, and the time-series was first linearly up-sampled from monthly to two-week intervals to increase the temporal resolution of the datasets before the time-series was smoothed using a Savitsky-Golay filter with a window length of 11 and a polynomial order of three. Though we report day-of-year as the unit for POS, the actual POS could have occurred anytime withing a given bi-monthly time step, so DOY values should be considered an approximation.

To avoid applying phenology trend analysis on regions that do not experience regular seasonal variation, we created a mask that removes regions identified as 'non-seasonal' using the definitions and methods defined by Moore et al. (2016). Broadly, the mask is created using three inputs: the standard deviation in NDVI anomalies, long-term mean NDVI, and the standard deviation in the mean seasonal cycle. These three inputs are used to identify regions that experience either low seasonal variability and low NDVI, or low seasonal variability and high interannual variability, which largely coincide with the desert bioclimatic region.

Per-pixel linear trends in these phenology metrics were extracted using the Theil-Sen robust regression approach, and significance was determined using a Mann Kendall test (significance defined α = 0.05). Trends summarised over bioclimatic regions were extracted by first calculating per-pixel robust

regression on the phenology statistics, and then summarising the trends within a bioclimatic region with kernel density estimation (KDE) plots.

## 3 Results

### 3.1 Quality of existing datasets.

The quality of the $NDVI_{AVHRR}$ products were compared against $NDVI_{MCD43A4}$ for the overlapping years 2000-2013. All datasets except $NDVI_{PKU-consolidated}$ perform poorly over regions with perennially high vegetation cover including wet coastal and highland forest ecosystems, where correlations between $NDVI_{AVHRR}$ and $NDVI_{MCD43A4}$ are close to zero in some regions (Fig. 3e-g). $NDVI_{CDR}$ and $NDVI_{GIMMS3g}$ also poorly represent the desert region with R values are as low as ~0.4 - 0.5. $NDVI_{PKU-consolidated}$ correlates very well with $NDVI_{MCD43A4}$ over most of the continent, with the exception of western Tasmania (Fig. 3h). Coefficients of variation are also high for the $NDVI_{GIMMS3g}$ and $NDVI_{PKU}$ datasets across much of the continent with average values of 0.33 and 0.18, respectively (Fig. 3b-c).

To demonstrate how the discrepancies over densely vegetated ecosystems would impact, Figure 3j-k presents a zonal timeseries of the woodlands of south-west Western Australia. These woodlands have been identified as a region of high endemic biodiversity (Myers et al., 2000; Hopper and Gioia, 2004), are vulnerable to the effects of long-term climate change, and are undergoing long-term shifts in climate (O'donnell et al., 2012; Hughes, 2011; Pitman et al., 2004; Hope et al., 2006). The MODIS-era interannual variability of these forests are shown through a rolling twelve-month mean timeseries (Fig. 3j) and reveal that all products capture interannual variability of the MODIS era reasonably well, though the long-term mean NDVI value varies substantially between products. The mean seasonal cycle, shown in Figure 3k (calculated from 2001-2013), reveals that the seasonal cycle of the forest ecosystem is very poorly represented in three of the four products, while $NDVI_{PKU-consolidated}$ tracks the overall shape of the seasonal cycle well. Discrepancies in seasonality are further highlighted in the per-pixel climatological 'month-of-maximum' NDVI plots (Fig. A3). Estimates of even this relatively straightforward metric of seasonality are impacted by the choice of dataset, with desert, savanna, and forested regions varying substantially between datasets, sometimes by as much as several months in the case of forested regions in Tasmania

and south-east Australia. The Australian-wide seasonal cycles likewise reveal substantial variation
between products (Fig. A3g).

To assess the quality of $NDVI_{AVHRR}$ products in the pre-MODIS era, Figure 4a compares the twelve-month rolling mean standardised anomalies of $NDVI_{Landsat}$ in the 1988-2000 period (based on a 1988-2012 climatology) with $NDVI_{AVHRR}$ anomalies. No product accurately tracks $NDVI_{Landsat}$ anomalies across the whole 1988-2000 period. Only the $NDVI_{PKU}$ product captures the amplitude of the La Niña driven positive anomaly of NDVI in 2000 (but recall that $NDVI_{PKU}$ is trained on the $NDVI_{Landsat}$ archive). In Australia, annual rainfall and NDVI anomalies are strongly correlated across the majority of Australia's land mass (Fig. 4c), demonstrating that vegetation growth across the continent is strongly water-limited (Peters et al., 2021; Poulter et al., 2014; Broich et al., 2014). It is therefore our expectation that similarly large negative and positive rainfall anomalies should result in similar NDVI anomalies in the pre-MODIS and MODIS eras. Taking the best of the products identified in the comparison with $NDVI_{MCD43A4}$, Figure 4b shows the twelve-month rolling mean standardised anomalies of $NDVI_{PKU-consolidated}$ from 1982-2022. In the MODIS era, $NDVI_{PKU-consolidated}$ responds strongly to anomalies in rainfall (background shading shows the continental average standardised rainfall anomalies), while in the pre-MODIS era significant droughts (e.g., 1982-83) and widespread rainfall events (e.g., 2000) produce comparatively little effect in NDVI, suggesting a lack of rainfall-driven variability over Australia in the pre-MODIS era. We develop the statistical relationships between annual mean standardised rainfall and NDVI anomalies, averaged across Australia, for the $NDVI_{MCD43A4}$ and $NDVI_{PKU-consolidated}$ products to quantify their sensitivity to water-supply. Considering the slope of the linear relationship between rainfall and NDVI to be an approximation of the sensitivity of NDVI to water supply, then $NDVI_{PKU-consolidated}$ in the 2000-2022 period displays a similar sensitivity (slope = 1.36, Fig. 4f) and correlation ($r^2=0.56$) as $NDVI_{MCD43A4}$ does in the same period (slope=1.13, $r^2=0.54$, Fig. 4d). Contrast this with $NDVI_{PKU-consolidated}$ in the 1982-2000 period where the apparent sensitivity is approximately half that of the 2000-2022 period (slope=0.65, Fig. 4e). While we may expect some changes in water-supply sensitivity over the decades due to effects such as $CO_2$

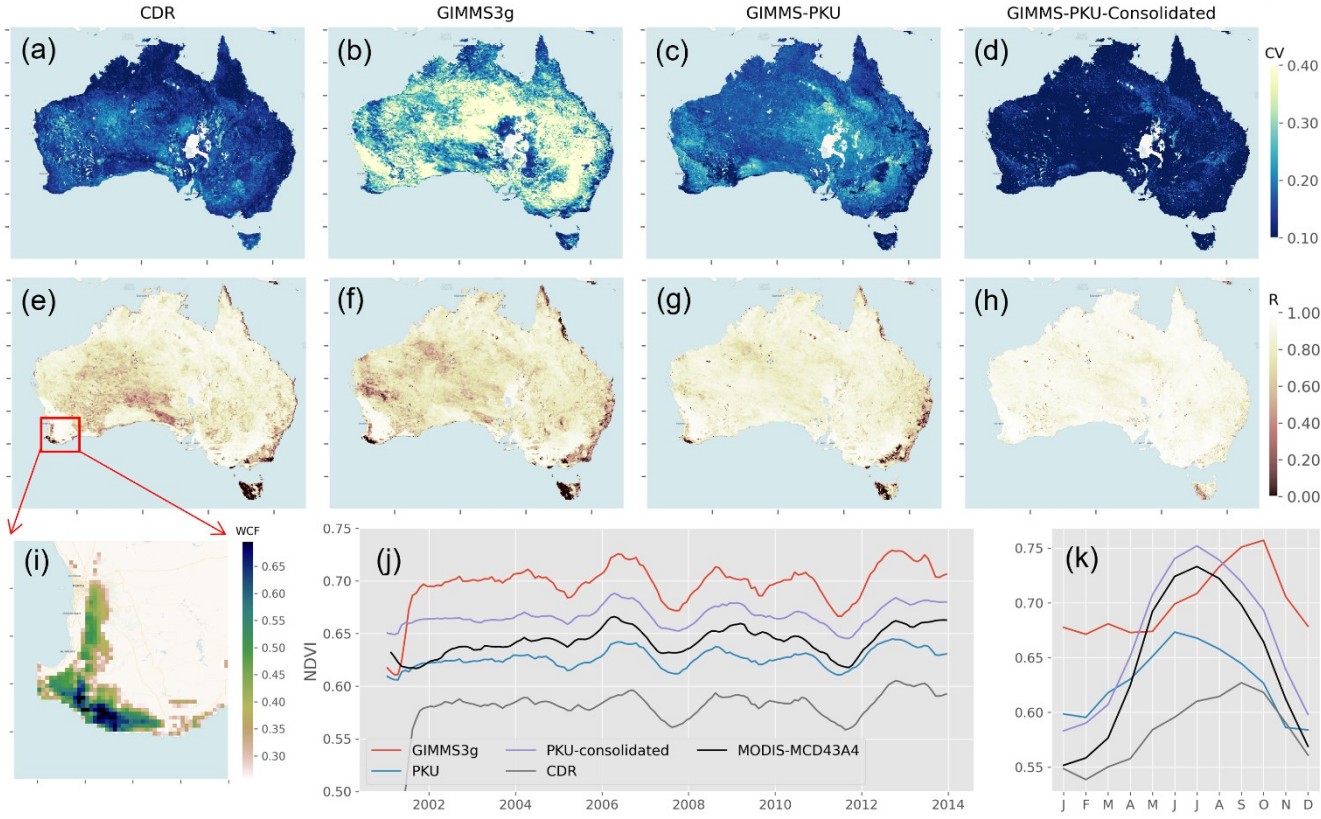

**Figure 3: Comparisons between NDVI$_{MCD43A4}$ and four versions of NDVI$_{AVHRR}$. a-d) The coefficient of variation (CV) between NDVI$_{MCD43A4}$ and NDVI$_{AVHRR}$ where RMSE is divided by the 2001-2013 mean of NDVI$_{MCD43A4}$. e-h) Pearson correlation (r) between NDVI$_{MCD43A4}$ and NDVI$_{AVHRR}$. i) Woody cover fraction (WCF) of the forests in south-west Western Australia indicating the location of the zonal time-series of (j) and (k). j) Twelve-month rolling mean NDVI timeseries of the forests of south-west Western Australia. k) Mean seasonal cycle of the forests of south-west Western Australia calculated over the 2001-2013 period.**

fertilisation (Donohue et al., 2013; Ukkola et al., 2016), a doubling of water-supply sensitivity is highly unlikely. Thus, we argue that no current NDVI$_{AVHRR}$ product currently satisfies our criteria of a product that both agrees well with NDVI$_{MCD43A4}$ while also producing satisfactory results in the pre-MODIS era.

### 3.2 Calibration and harmonisation performance

Independent validation statistics for all six model varieties ('clim' and 'noclim'; desert, high and low WCF) reveal a high degree of agreement in all model types with $r^2 \geq 0.91$ for the 'clim' models, RMSE $\leq 0.039$, and MAE $\leq 0.028$ (Fig. 5a-c). The 'clim' model types tended to have errors ~15 % smaller

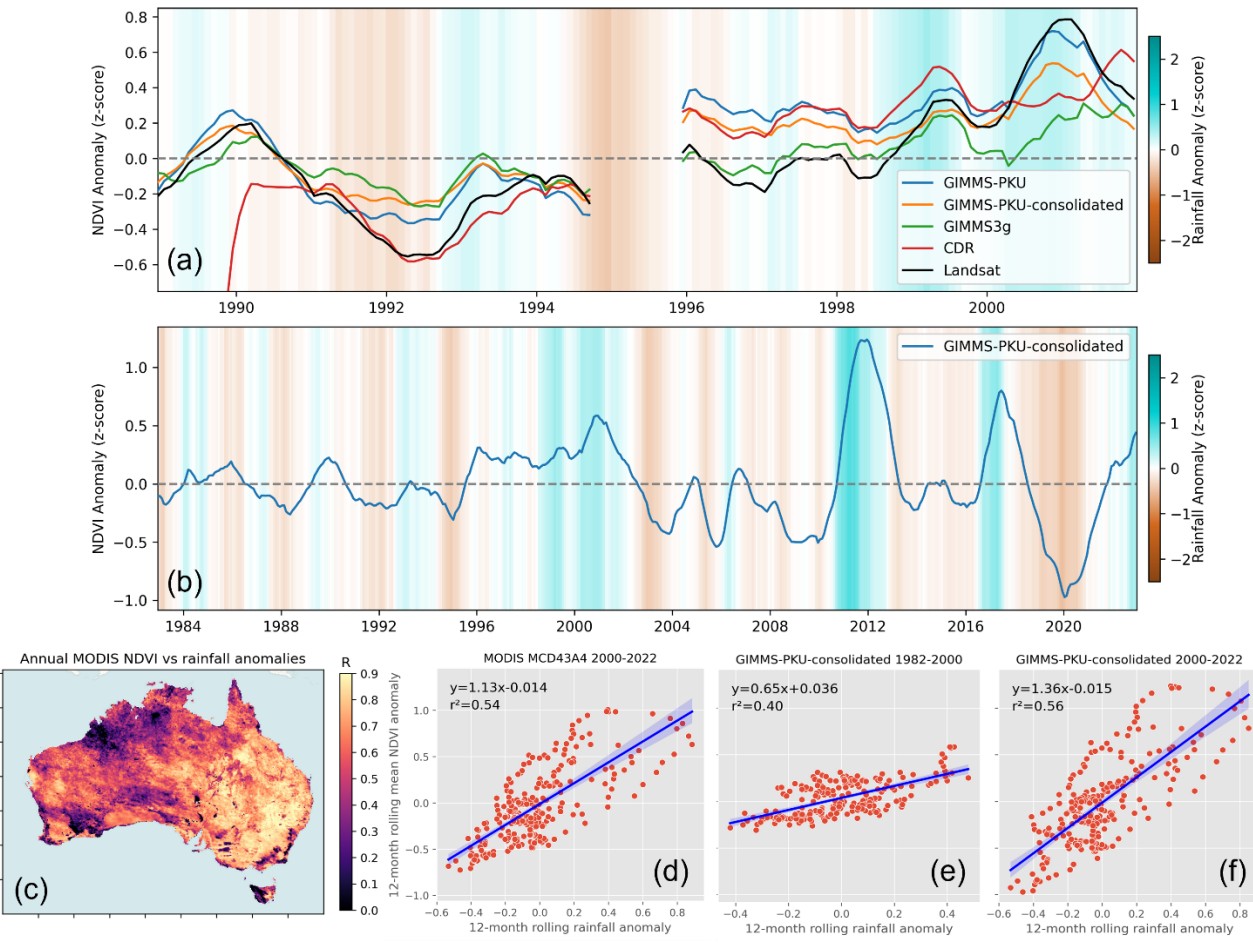

**Figure 4: a) Twelve-month rolling mean standardised anomalies of Landsat, CDR, GIMMS3g, GIMMS-PKU, and GIMMS-PKU-consolidated NDVI, based on a common 1988-2012 climatology. Background shading represents twelve-month rolling mean standardised rainfall anomalies. All datasets, besides rainfall, have matching data gaps. b) Twelve-month rolling mean standardised anomalies of the NDVI$_{PKU-consolidated}$ product (1982-2022 climatology). c) Pearson correlations between annual NDVI$_{MCD43A4}$**
**anomalies and annual rainfall anomalies, shown here to demonstrate the strongly water limited nature of Australia's vegetation. d-f) Relationships between twelve-month standardised rainfall and NDVI anomalies averaged across Australia for different periods and different products. In (d) NDVI$_{MCD43A4}$ and rainfall anomalies have been calculated against a 2000-2022 baseline. In (e-f) rainfall and NDVI$_{PKU-consolidated}$ anomalies have been calculated against a 1982-2022 baseline. The relationships y=mx+c denotes the linear regression slope between rainfall and NDVI anomalies where y is NDVI anomalies, x is rainfall anomalies, and m is the slope**
**coefficient. The slope coefficient can be considered an approximation of the sensitivity of NDVI to anomalous water supply aggregated over the continent.**

than their 'noclim' counterparts (Fig. 5d-f). SHAP feature importance plots indicate NDVI$_{CDR}$ as the

most important variable (Figure A3), but in the high WCF regions the relative importance of NDVI$_{CDR}$

diminished and NDVI$_{MCD43A4}$ summary statistics, solar radiation, and cumulative rainfall substantially

impacting predictions (Figure A4b,c).

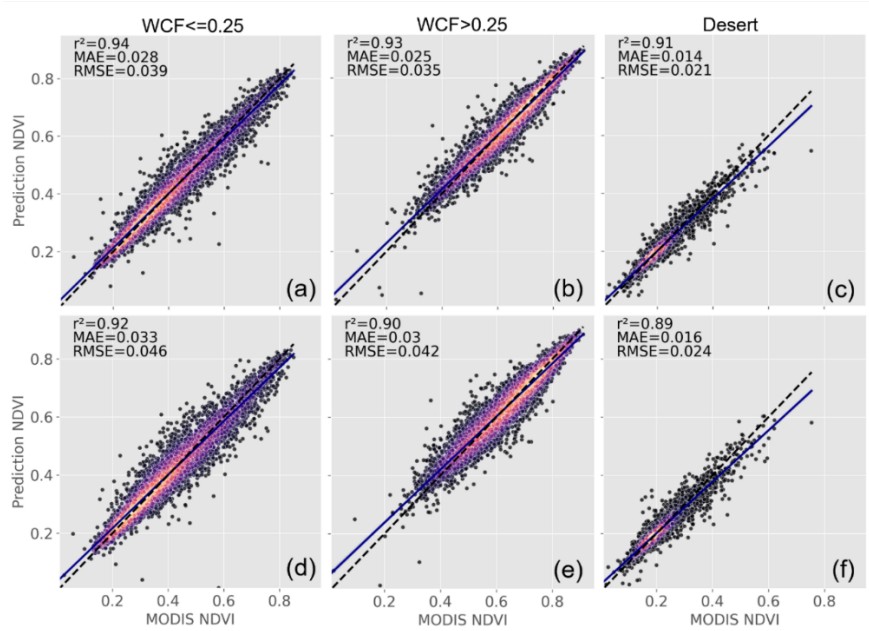

**Figure 5: Validation scatter plots for the calibration and harmonisation between NDVI_CDR and NDVI_MCD43A4. (a-c) show the results for the 'clim' model. (d-f) shows the same but for the 'noclim' model type.**

Per pixel agreements between NDVI_AusE and NDVI_MCD43A4 for both the 'clim' and 'noclim' model types reveal a very high degree of correlation across the continent (note that pixels with a long-term average NDVI ≤ 0.11 are masked for this analysis). Correlations between NDVI_MCD43A4 and NDVI_AusE in Australia's forested ecosystems have been greatly improved, averaging Pearson R = 0.85 (Fig. 6a) in the 'clim' model (average Pearson R in the CDR = 0.48). Areas of lower correlation persist in places that experience ephemeral or periodic water inundation such as mangroves and inland lake systems. Relative error has been reduced universally across the continent, with a continental average CV of <10 % (Fig. 6b). Areas of greatest relative error occur in the channel country in Australia's arid interior, and the irrigated regions of the northern Murray Darling Basin. The 'noclim' model performs similarly, though correlations and relative error are universally worse than the 'clim' model (Fig. 6c-d). Residual NDVI values after subtracting NDVI_AVHRR from NDVI_MCD43A4 before and after the calibration and harmonisation show the GBM model has entirely removed the residual seasonal signal present in the CDR product, resulting in residuals that closely track the zero line. Some small bias remains in the 2011-2012 period (particularly for the 'noclim' model) when anomalously large rainfall related to a major La Nina

event resulted in anomalous greening in the savanna and desert biomes. This is further illustrated in Figure A5 where NDVI timeseries before and after the adjustment have been summarised over six bioclimatic regions (extents in Fig. 2b). Differences in the Australia-wide time-series between $NDVI_{MCD43A4}$ and $NDVI_{AusE}$ are largely attributable to $NDVI_{AusE}$ underestimating peak NDVI during 2011-2012 in the desert and savanna biomes (Fig. A5f-g).

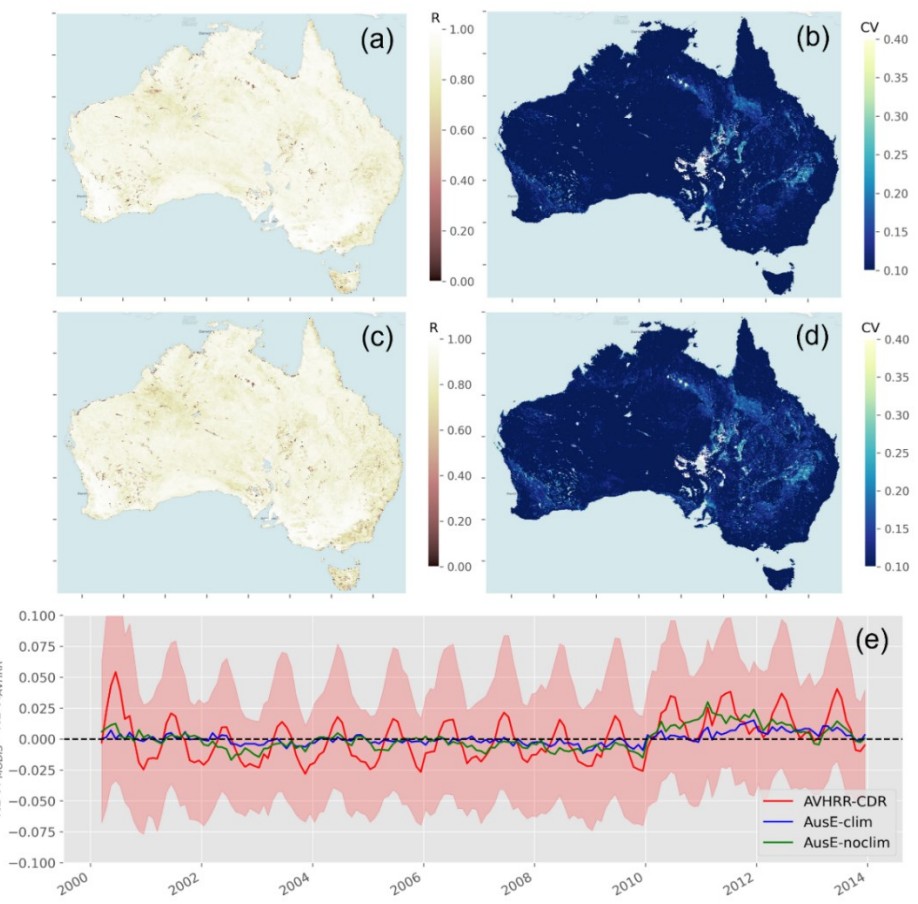

**Figure 6: Results of the calibration and harmonization between $NDVI_{CDR}$ and $NDVI_{MCD43A4}$. a) shows the per pixel Pearson correlation, between $NDVI_{MCD43A4}$ and 'clim' $NDVI_{AusE}$. b) shows the same as (a) but for the coefficient of variation. c-d) the same as (a-b) but for the 'noclim' model type. e) The residual NDVI value when subtracting $NDVI_{AVHRR}$ from $NDVI_{MCD43A4}$ before and after the calibration and harmonization. Residuals are calculated per pixel and then averaged over Australia. Shading indicates the**
370 **standard deviation in residuals across the continent for the $NDVI_{CDR}$ product.**

Improvements in the alignment between $NDVI_{CDR}$ and $NDVI_{MCD43A4}$ from this regional calibration and harmonisation are further demonstrated in Figure 7 where timeseries are summarised over two challenging forest ecosystems in southwest Western Australia and Tasmania. Mean seasonal cycles

between the two NDVI datasets are now in very close agreement (Fig. 7c, f) and the NDVI$_{\text{AusE-clim}}$ time-series from 1982-2000 can effectively integrate with the NDVI$_{\text{MCD43A4}}$ time-series without introducing major discontinuities (Fig. 7b, e). Note also that the GBM calibration has ameliorated the strong increasing trend in NDVI$_{\text{CDR}}$ from 1982-2000 (Fig. 7b, e) that is almost certainly due to artificial step changes between sensor transitions and poor calibration over these regions. In the appendix, we replot Figure 7d-f with the inclusion of NDVI$_{\text{GIMMS3g}}$ to demonstrate that the trend in NDVI$_{\text{CDR}}$ is an artefact of the CDR product (Fig. A6).

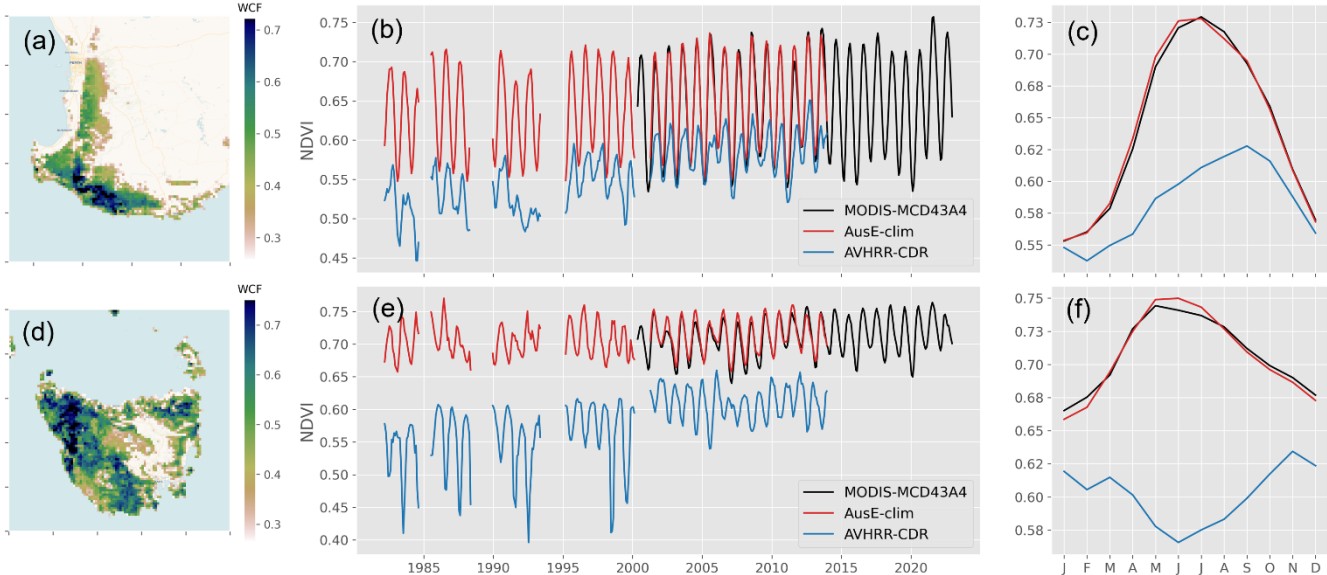

Figure 7: Results before and after the calibration and harmonisation of NDVI$_{\text{CDR}}$ for two example high woody canopy cover regions previously identified as having the worst agreement with NDVI$_{\text{MCD43A4}}$. b-c) Three-month rolling mean 1982-2022 NDVI time series, and the mean seasonal cycle (averaged over the 2001-2013 period), respectively, for the forests of south-west Western Australia. e-f) Same as (b-c) but for Tasmanian forests. Time series are the spatial average of the regions to their left.

## 3.2 Gap-filling with Synthetic NDVI

The NDVI$_{\text{SYN}}$ dataset record agrees exceptionally well with the joined NDVI$_{\text{AusE-clim}}$ and NDVI$_{\text{MCD43A4}}$ series when aggregated across Australia (Fig. 8e). The time series of Figure 8e is further disaggregated into high and low WCF regions (as per Figure 2a) in Figure A7 and reveals that in densely wooded regions synthetic NDVI tends to underestimate peak seasonal growth, but otherwise captures seasonal timings and inter-annual variability (Fig. A7b). In the low WCF regions (Fig. A7a), synthetic NDVI closely matches observations. At the pixel level, the long-term mean NDVI of both datasets is virtually identical

(Fig. 8a-b). Per-pixel Pearson correlation averages 0.85 across the continent (Fig. 8d). Areas of poorer correlation occur in western Tasmania, the highlands forests of south-east Australia – all areas that experience seasonal snow fall – and regions of either anthropogenic water application (irrigation) or ephemeral, delayed water inundation (inland rivers in the arid interior). Mean relative error was also low, averaging 11 %, but with hotspots of greater error again occurring in the regions where water inundation is not dependent on direct rainfall (Fig. 8c) The results before and after gap filling $NDVI_{AusE-clim}$ are presented in Figure 8f. As missing data tends to be in the higher NDVI regions (wetter, cloudier, forested regions), gap filling has the tendency of increasing NDVI when averaged over the continent.

We present validation scatter plots and feature importance plots for the desert and non-desert GBM models in the appendix (Fig. A8). In the non-desert region, three-month cumulative rainfall and VPD are the key climate drivers of predictions, while in the desert region, six-month cumulative rainfall, VPD, and incoming solar radiation are the key climate drivers.

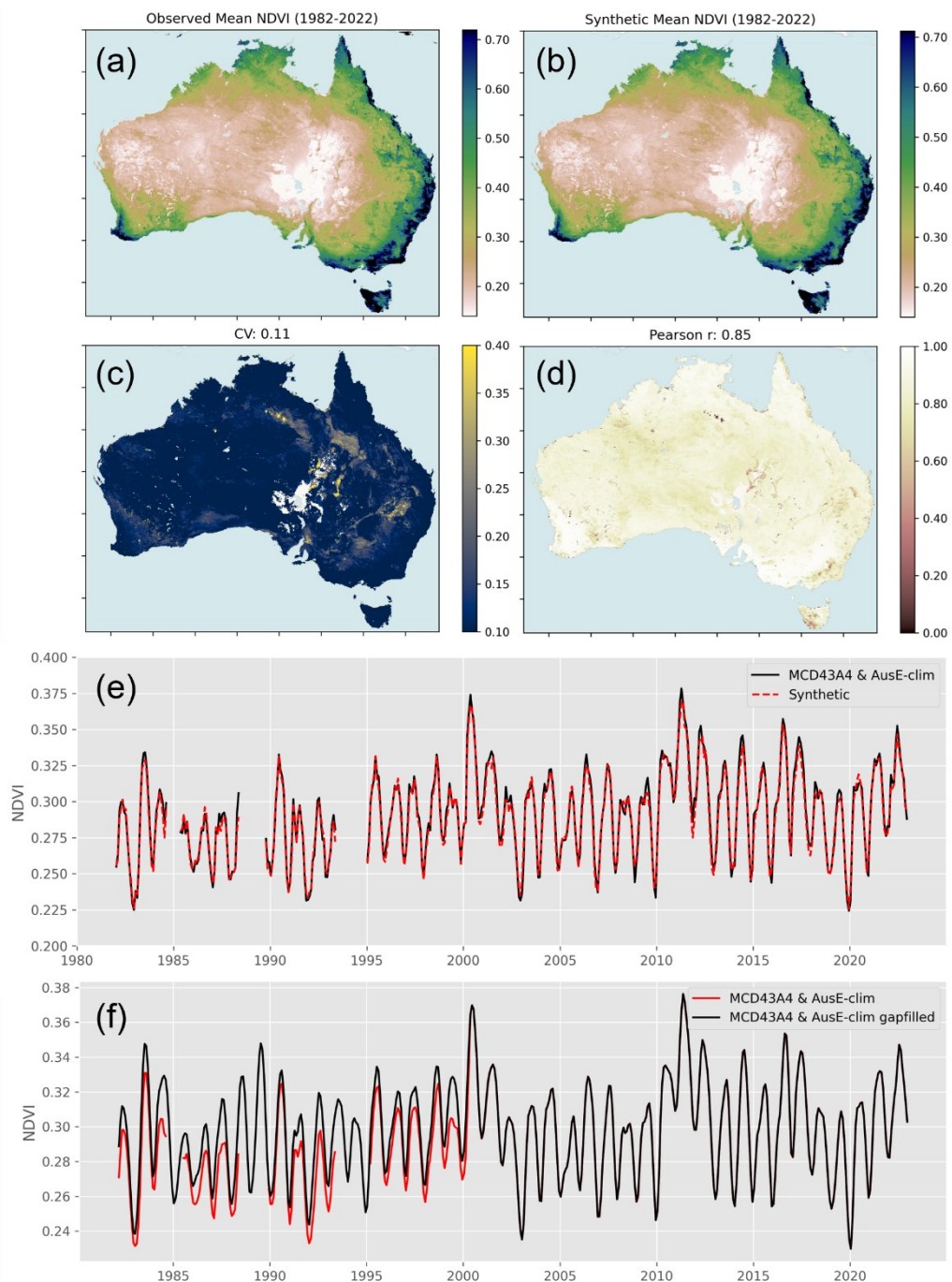

**Figure 8: Evaluation of the synthetic NDVI built to gap-fill the NDVI_AusE-clim record a-b) show the observed and synthetic long-term mean NDVI, respectively. c) per pixel coefficient of variation (CV) between observed NDVI and synthetic NDVI. d) Same as (c) but Pearson correlation. e) Continentally averaged observed and synthetic NDVI timeseries, where data gaps have been matched. f) The results of gap filling the merged NDVI_AusE-clim and NDVI_MCD43A4 time series.**

## 3.3 Assessing interannual variability

Comparing the calibrated, harmonised, and gap-filled $NDVI_{AusE-clim}$ dataset with rolling annual mean $NDVI_{Landsat}$ anomalies reveals a good level of agreement in both the timing and magnitude of inter-annual variability throughout the 1988-2012 period (Fig. 9a). $NDVI_{PKU-consolidated}$ is also shown for comparison and gaps in the $NDVI_{PKU-consolidated}$ dataset have been filled using the same synthetic data and procedure as $NDVI_{AusE-clim}$ to facilitate a more straightforward comparison and continuous time-series. $NDVI_{AusE-clim}$ consistently outperforms $NDVI_{PKU-consolidated}$ throughout the Landsat series. IAV in $NDVI_{AusE-clim}$ is further assessed in Figure 9b where the full time series (1982-2022, joined with $NDVI_{MCD43A4}$) and $NDVI_{PKU-consolidated}$ are plotted together as rolling annual mean standardised anomalies against the same 1982-2022 climatology. $NDVI_{AusE-clim}$ clearly displays greater IAV in the pre-MODIS era. We repeat the same analysis as in Figure 3d-f but this time including $NDVI_{AusE-clim}$. The NDVI-rainfall relationships show that $NDVI_{AusE-clim}$ reports a similar water-supply sensitivity and correlation in the 1982-2000 period (slope=1.28, $r^2$=0.51, Fig. 8d) as MODIS does in 2000-2022 period (slope=1.13, $r^2$=0.54, Fig. 9c). Again, while we may expect some changes in water-supply sensitivity over the decades due to effects such as $CO_2$ fertilisation, water supply sensitivity ought to remain relatively stationary, and we take the correspondence between $NDVI_{MCD43A4}$ sensitivity and $NDVI_{AusE-clim}$ sensitivity as an indication that $NDVI_{AusE-clim}$ is responding realistically to interannual variations in rainfall.

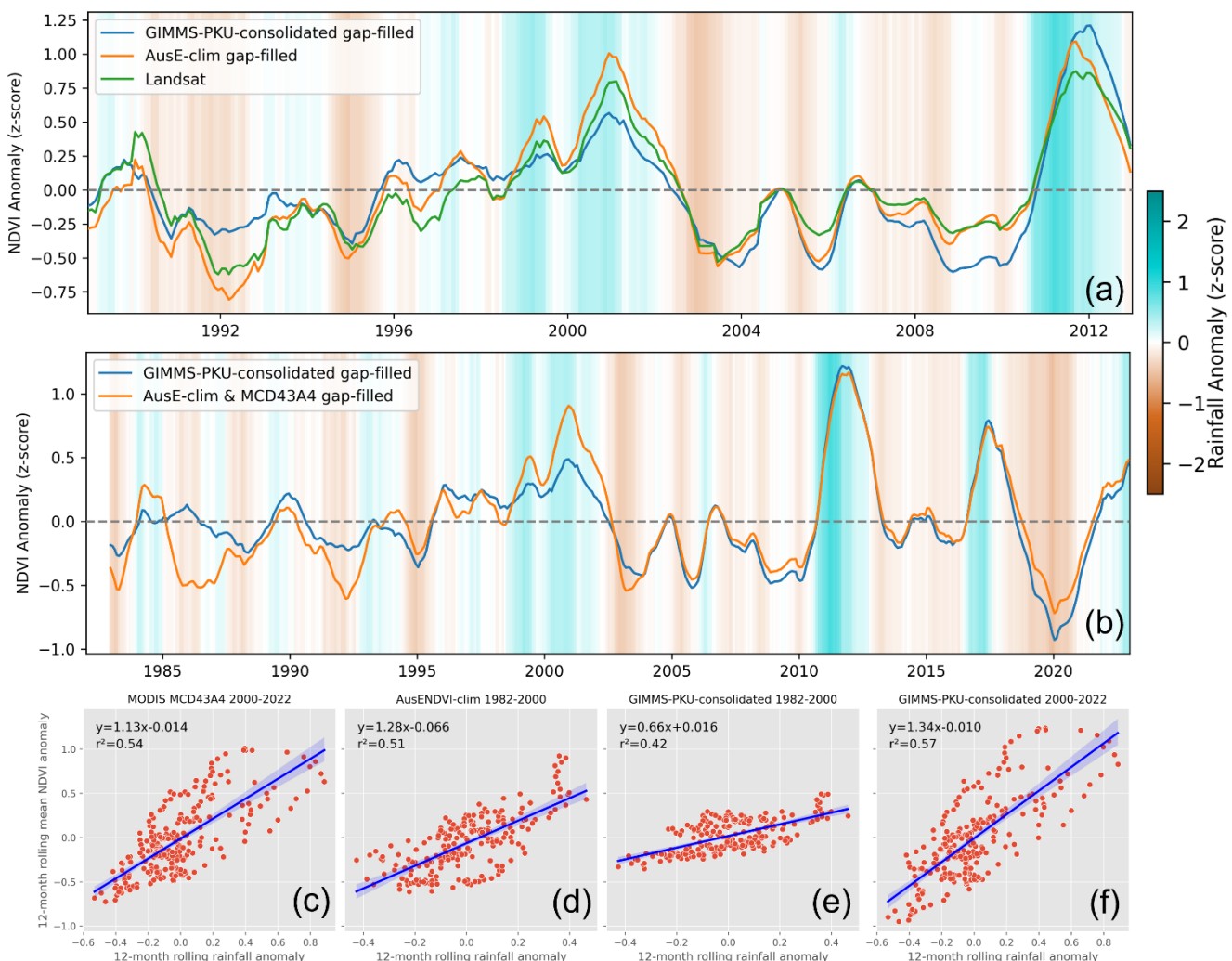

**Figure 9: a)** Twelve-month rolling mean standardised NDVI anomalies of the gap-filled NDVI$_{AusE-clim}$ plotted alongside Landsat anomalies and NDVI$_{PKU-consolidated}$ anomalies. Gaps in the NDVI$_{PKU-consolidated}$ dataset have been filled using the same synthetic data and procedure as NDVI$_{AusE-clim}$. All datasets are matched to Landsat data gaps. **b)** Twelve-month rolling mean standardised anomalies of the NDVI$_{PKU-consolidated}$ (gap-filled in the same manner as (a)), and NDVI$_{AusE-clim}$ joined with NDVI$_{MCD43A4}$ (1982-2022 climatology). **c-f)** Relationships between twelve-month standardised rainfall and NDVI anomalies averaged across Australia for different periods and different products. Rainfall, NDVI$_{AusE-clim}$ and NDVI$_{PKU-consolidated}$ anomalies have been calculated against a 1982-2022 baseline. NDVI$_{MCD43A4}$ anomalies have been calculated against a 2000-2022 baseline. The slope coefficient can be considered an approximation of the sensitivity of NDVI to anomalous water supply aggregated over the continent. Note that the slope and intercepts for GIMMS-PKU-consolidated are slightly different to Figure 3 owing to gap filling.

## 3.4 Annual average trends

We also evaluated the annual-average NDVI trends across Australia to assess the performance of AusENDVI in reproducing greening trends observed in other products. Trends were calculated over the overlapping period of 1982-2013 using ordinary least squares regression after aggregating NDVI data to annual means. AusENDVI closely reproduces the observable trends in $NDVI_{GIMMS3g}$ (coefficients: AusENDVI-clim=0.00056 NDVI yr$^{-1}$, AusENDVI-noclim=0.00049 NDVI yr$^{-1}$, GIMMS3g=0.00062 NDVI yr$^{-1}$; Fig. 10). Trends in $NDVI_{MCD43A4}$ over the shorter interval from 2000-2013 displayed a similar slope to AusENDVI and GIMMS3g (0.00051 NDVI yr$^{-1}$). Trends in the two GIMMS-PKU products are approximately half those of the other products.

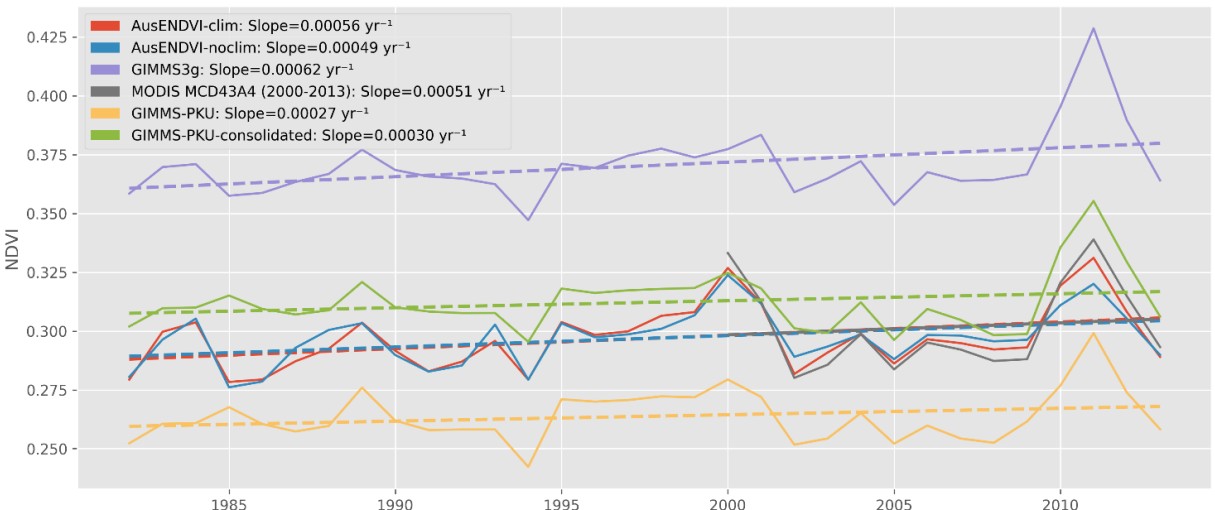

**Figure 10: Annual average NDVI trends summarised over Australia for the overlapping period of 1982-2013. All data gaps have been matched between datasets and datasets have been reprojected to match the resolution of GIMMS3g. Note that AusENDVI-clim and noclim have both had data gaps filled to facilitate better annual averaging (i.e., so all years have values). Trend lines have been fitted using ordinary least-squares regression and coefficients are expressed in terms of NDVI per year.**

## 3.5 Trends in peak-of-season phenology

Per-pixel trends in vPOS, POS and the 40-year median values for these statistics are shown in Figure 11. Trends in vPOS are almost universally positive across the continent (hatching indicates a significant trend), with the exceptions of inland northern Murray-Darling Basin, the eastern periphery of the wheat

belt in Western Australia, and the region north of Adelaide (Fig. 11b). Positive trends observed in the major agricultural region of the Murray-Darling Basin and the northern half of the West Australian wheat belt and are non-significant. Distributions of trends in vPOS, stratified by bioclimatic region, reveal the highest median trends are recorded in the tropics and savanna regions at 0.0013 and 0.0014 NDVI yr$^{-1}$, respectively (Fig. A9a-e). The Mediterranean region has the lowest median trend at 0.0009 NDVI yr$^{-1}$.

Trends in the day-of-year that peak NDVI occurs (POS) are negative across much of the continent, suggesting there is a general tendency for NDVI to peak earlier in the year across Australia. Significant negative trends occur in the agricultural zones of the Mediterranean bioclimatic region, the greater western woodlands that border the eastern margin of the WA wheatbelt, the western half of the Nullabor plain, parts of the Riverina agricultural region of south-western New South Wales and extending into Victoria, and western parts of the northern tropical savanna. These significant negative trends are reflected in the POS trend distributions in Figure A9f-j where the median trend in the warm temperate and Mediterranean regions are highest at 3.4 and 2.3 days per decade, respectively. Significant positive trends (peak NDVI occurring later in the year) are observed in tropical northern Queensland and western Tasmania and can be as high as 5-10 days per decade.

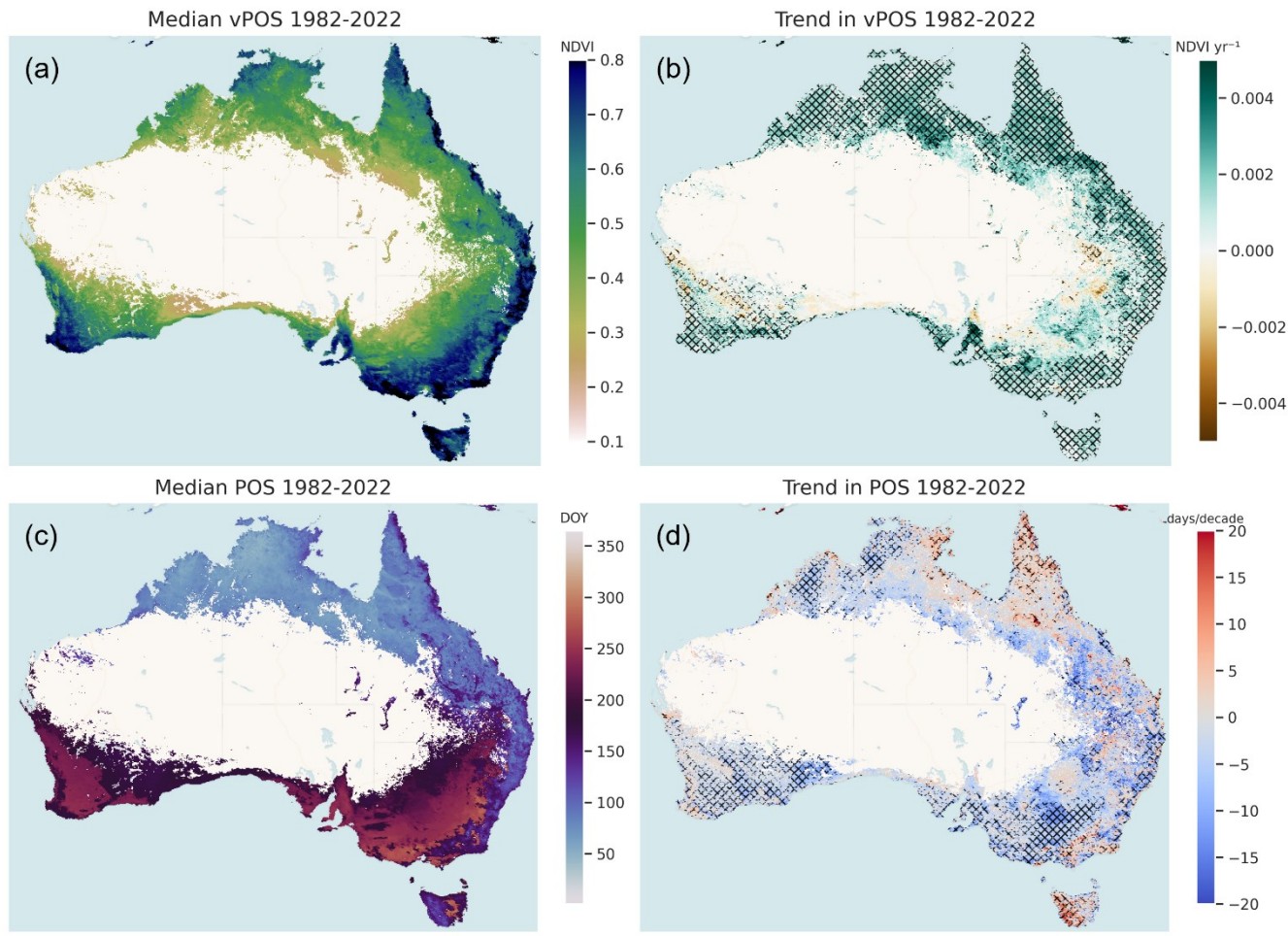

**Figure 11: a) The median annual peak NDVI value (vPOS) from 1982-2022. b) Theil-Sen robust regression trends in vPOS. c) Median day-of-year that peak NDVI occurs (POS), 1982-2022. d) Theil-Sen robust regression trends in POS. Hatching on trend plots indicates significance at alpha=0.05 using a Man-Kendall test. All plots are derived from the gap-filled 'clim' NDVI$_{AusE}$ dataset. Non-seasonal areas have been masked using the method described in section 2.4.**

## 4    Discussion

### 4.1 Limitations of existing global products and improvements by AusENDVI

We expected to identify differences between NDVI$_{AVHRR}$ and NDVI$_{MCD43A4}$ given differences in the spectral sampling between sensors, their different pre-processing and atmospheric corrections methods, spatial resolutions and temporal compositing techniques. Likewise, comparatively lower correlations in the densely vegetated regions were also expected due to the total variance in evergreen forests being

smaller than for seasonal vegetation (grassland, croplands), and therefore, assuming a similar unexplained variance (noise), correlations should necessarily be weaker. Nonetheless, we were surprised by the fairly large inconsistencies between $NDVI_{GIMMS3g}$, $NDVI_{CDR}$, and $NDVI_{GIMMS-PKU}$ in representing the seasonal dynamics of Australia's densely vegetated regions (e.g. Fig. 3k). Why this is the case deserves a greater focus of study than we devote here but is likely related to some combination of the presence/absence of BRDF and water-vapour corrections, varying contamination by clouds, and any gap-filling procedures applied. Regardless of the reasons why, the intercomparison between $NDVI_{AVHRR}$ products highlights that global datasets, while often performing adequately when statistics are aggregated at the global or continental scale, can mask disparities that are important at the regional to local scale (Meyer and Pebesma, 2022). We advocate closely examining regional and local contexts to assess how suitable a given NDVI dataset is for a particular use case. For example, in Australia seasonal cycles in $NDVI_{CDR}$ are highly suspect and thus should not be relied upon for phenology studies. However, $NDVI_{CDR}$ has a comparatively low relative error when compared with $NDVI_{MCD43A4}$ and displays reasonable inter-annual variability so would likely be more suited to long-term studies of agricultural drought frequency or the impacts of $CO_2$ fertilisation on canopy cover (assuming sensor transitions are filtered). In Australia, the best use of $NDVI_{PKU-consolidated}$ is likely the reverse, its representation of seasonal cycles comports well with $NDVI_{MCD43A4}$, while IAV is subdued in the pre-MODIS era which could lead to incorrect conclusions regarding shifting sensitivities to water supply in Australia's water-limited ecosystems. In general, we urge caution in using existing global $NDVI_{AVHRR}$ products for studying vegetation trends and seasonality in Australia. AusENDVI shows significant improvement over existing global datasets in this respect. The improved correspondence in seasonal cycles between AusENDVI and $NDVI_{MCD43A4}$ provides evidence that AusENDVI is more suitable for exploring longer-term changes to Australia's vegetation phenology. Moreover, the addition of climate features to the calibration and harmonisation also appears to have improved the representation of long-term interannual variability and trends in annual average NDVI, thus AusENDVI-clim should likewise offer a better basis for studying the shifting frequency of extreme climate events and their impact on the terrestrial biosphere.

## 4.2 Synthetic NDVI

The creation of a synthetic NDVI using only climate, $CO_2$ concentration, and woody cover fraction as predictors revealed a high degree of predictability in NDVI over much of Australia. Regions of lower predictability were located where water supply is either from elsewhere or delayed (ephemeral inland rivers) or from irrigation. In the absence of features that could describe water supply without rainfall, NDVI patterns in these zones will continue to be difficult to estimate if direct satellite observations are unavailable. Notwithstanding some spatial variability in per-pixel predictability, in general the high degree of agreement between observed and synthetic NDVI presents the prospect of extending the synthetic NDVI further back in time through the observational climate record, which in Australia is reliable throughout much of the 20[th] century. In land surface models, a dynamic phenology algorithm is an important sub-model which influences the overall carbon cycle, evapotranspiration, and energy balance of the model (Chen, 2022). The long-term record of synthetic NDVI developed here could, therefore, prove useful for validating the development of process-based phenology models for Australia's diverse range of vegetation and climate. Or, with empirically validated NDVI-LAI relationships, AusENDVI could be used as a phenology forcing during the pre-satellite era for the many LSMs that do not dynamically simulate LAI.

## 4.3 Sources of uncertainty and future work

There are several sources of uncertainty in AusENDVI. Firstly, the climate and landscape features used are subject to their own uncertainties which will undoubtedly propagate into both the calibration and harmonisation, as well as the gap-filling with synthetic NDVI. For example, rainfall station observations in the arid interior of Australia are relatively sparse so errors in the spatial interpolation of rainfall are highly likely. Uncertainties in the $NDVI_{CDR}$ product are also likely to be transmitted to our dataset. Future work may include a greater treatment of uncertainty through ensemble modelling where climate features (e.g., different rainfall and solar radiation datasets), and model types used for fitting are iterated to generate an uncertainty envelope. We also aim to assess how well NDVI from the Visible Infrared Imaging Radiometer Suite (VIIRS) agrees with $NDVI_{AusE}$ and $NDVI_{MCD43A4}$ over Australia. Should there be a substantial discrepancy, the methods described here could be applied to VIIRS to create an ongoing,

updated NDVI dataset for Australia than can continue to form the foundation for continental-scale studies of terrestrial ecosystem change. Irrespective, we argue our AusENDVI estimates are based on the best available data, while the gradient boosting models have gone through extensive cross-validation. Therefore, we contend that the resulting trends should be more accurate than any alternative NDVI dataset.

## 4.4 Trends in peak of season phenology

We identified advances in the timing of POS across much of Australia's land mass (though not all). Over the Mediterranean, warm temperate, and cool temperate bioclimatic regions the median peak phenology trends were -2 to -3 days/decade. Advances in plant maturity in the southern hemisphere from field data are also reported by Chambers et al. (2013) where the mean rate of change in plant maturity was 14 days/decade, mostly from temperate regions (63 % of their data are from grape-vines). This rate of change is comparable to the per-pixel rates of change in POS that are seen in parts of the Mediterranean and warm temperate regions where it is not uncommon to see negative trends ranging from 10-15 days/decade (Fig. 11d). However, the magnitude of a trend is influenced by the length of the time series so comparisons with variable length field data is difficult and shorter records are more likely to report a larger rate (Chambers et al., 2013). Advances in the timing of POS could be due to a combination of climate drivers. In the northern hemisphere, warming has led to earlier peak greening (Huang et al., 2023; Liu et al., 2021; Park et al., 2019). Warming can accelerate metabolism, so where water is non-limiting, leaf development can be faster. However, temperature increases also increase vapour pressure deficits which decrease water-use efficiency and can reduce plant productivity, though this effect may be compensated for by enhanced $CO_2$ (Rifai et al., 2022; Dusenge et al., 2019). Changes in the timing of peak rainfall may also contribute to shifts in the timing of peak NDVI. The timing of peak climatological rainfall has shifted since 1960 (Fig. A10a-c), and there is some coincidence between trends in POS and shifts in rainfall POS (e.g., advancement around Adelaide). The goal of this study is not to draw conclusions on the likely drivers of seasonality change in Australia, but to argue that our dataset provides a more reliable means for tackling these questions. Future work will delve into a greater suite of phenology metrics (e.g., start-

of-season, end-of-season, growing season length (Xie et al., 2023)), and explore the drivers of phenological change.

The pervasive positive trends in vPOS are consistent with results elsewhere and are likely due to the impacts of $CO_2$ fertilisation, which allows a given amount of precipitation to sustain a greater maximum level of plant production over time (Donohue et al., 2009; Donohue et al., 2013; Rifai et al., 2022; Ukkola et al., 2016). Increases in the magnitude of Austral spring and summer rainfall in northern Australia are also likely to have contributed to the widespread increase in vPOS in tropical Australia (Figure A10d). It is also likely that improving agricultural practices has increased maximum NDVI in the rain-fed cropping regions, especially in South Australia and Victoria where positive vPOS trends are significant. Trends in maximum NDVI in the WA wheatbelt are also positive, but contrast with the fact that WA has seen a widespread autumn drying trend (Fig. A10d). We speculate that agricultural innovation here has counteracted a drying trend that would otherwise have reduced foliage cover.

## 5 Data and Code Availability

AusENDVI is openly available at https://doi.org/10.5281/zenodo.10802704 (Burton, 2024) and consists of several datasets. Each dataset has a description in the attributes of the NetCDF file that defines its provenance. A short description of each dataset is provided below as an additional reference. All datasets are in "EPSG:4326" projection, have a spatial resolution of 0.05˚, and monthly temporal resolution. A Jupyter notebook is also provided at the above link demonstrating how to load, plot, mask, reproject, and gap-fill AusENDVI datasets.

- *AusENDVI-clim_1982_2013*. Calibrated and harmonised NOAA's Climate Data Record AVHRR NDVI data from January 1982 to December 2013. This version of the dataset used climate data in the calibration and harmonisation process. The dataset has not been gap filled, and extra data has been filtered/removed beyond the typical QA filtering using methods described in this publication.
- *AusENDVI-noclim_1982_2013*. Calibrated and harmonised NOAA's Climate Data Record AVHRR NDVI data from January 1982 to December 2013. This version of the dataset did not use

climate data in the calibration and harmonisation process. The dataset has not been gap filled, and extra data has been filtered/removed beyond the typical QA filtering using methods described in this publication.

- *AusENDVI-synthetic_1982_2022*. This dataset consists of synthetic NDVI data that was built by training a model on the joined 'AusENDVI-clim' and 'MODIS-MCD43A4 NDVI' timeseries using climate, woody-cover-fraction, and atmospheric $CO_2$ as predictors.

- *AusENDVI-clim_gapfilled_MCD43A4_1982_2022*. This dataset consists of calibrated and harmonised NOAA's Climate Data Record AVHRR NDVI data from January 1982 to February 2000, joined with MODIS-MCD43A4 NDVI data from March 2000 to December 2022. This version of the dataset used climate data in the calibration and harmonisation process. The dataset has been gap filled using AusENDVI-synthetic,

- *AusENDVI-noclim_MCD43A4_1982_2022*. This dataset consists of calibrated and harmonised NOAA's Climate Data Record AVHRR NDVI data from January 1982 to February 2000, and MODIS-MCD43A4 NDVI data from Mar. 2000 to Dec. 2022. This version of the dataset did not use climate data in the calibration and harmonisation process. The dataset has not been gap filled.

The code to conduct all analysis described here is available on the open-source repository: https://github.com/cbur24/AusENDVI

# 6 Conclusion

We calibrated and harmonised $NDVI_{CDR}$ to $NDVI_{MCD43A4}$ for Australia using a well cross-validated gradient-boosting ensemble decision tree method. We developed two versions of the datasets, one that utilises climate data in the feature set to achieve the best possible agreement between $NDVI_{CDR}$ and $NDVI_{MCD43A4}$ ('AusENDVI-clim'); and a second dataset that does not rely on climate data ('AusENDVI-noclim'). The resulting datasets have a spatial resolution of 0.05°and extend from 1982-2013 with a monthly time step. We also provide a complete 41-year long dataset where gap filled AusENDVI-clim from January 1982 to February 2000 is seamlessly joined with $NDVI_{MCD43A4}$ from March 2000 to

December 2022. The advantages of AusENDVI are that: 1) It closely reproduces the $NDVI_{MCD43A4}$ record in terms of seasonality, interannual variability, and trends in annual-average NDVI; 2) It reproduces annual anomalies in the Landsat NDVI record in the pre-MODIS era (back to 1988), and shows realistic rainfall-driven interannual variability back to 1982; 3) We developed a reliable method for gap filling the

AusENDVI record by creating a synthetic NDVI dataset using only climate, $CO_2$ concentration, and woody cover fraction as predictors. The resulting dataset showed excellent agreement with the observations, providing confidence in its use for gap filling. 4) AusENDVI has a higher spatial resolution than any of the GIMMS-based datasets and is built using inputs that apply the full suite of atmospheric and BRDF corrections; and 5) The methods and code for its development are entirely open-source. No

other existing product can lay claim to all these attributes which is why we argue AusENDVI is an important addition to the suite of NDVI products available. We contend it is highly suitable for studying the impact of global environmental change on Australia's terrestrial vegetation.

## Appendix

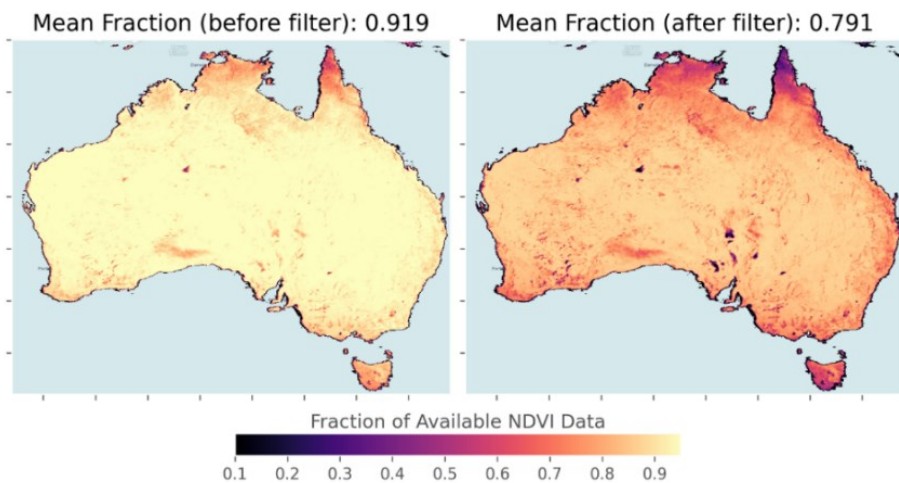

**Figure A1: Available fractions of data before and after additional filtering of $NDVI_{CDR}$ data. A value of one means all monthly time-steps between 1982-2013 are preserved.**

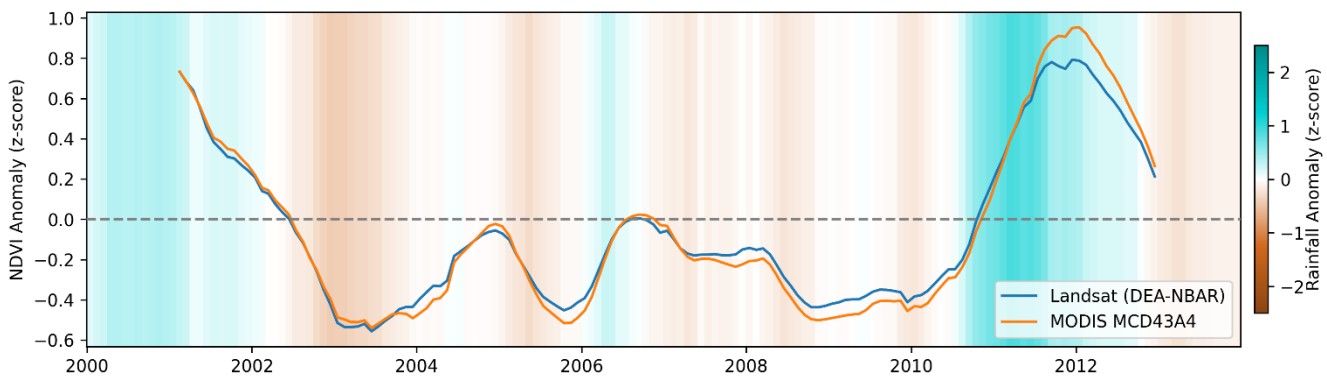

**Figure A2: Standardised anomalies of the overlapping period between MODIS MCD43A4 NDVI and DEA's Landsat NDVI derived from the common baseline period of 2000-2012. Rainfall anomalies are derived from a longer baseline of 1982-2022.**

**Table A1. The hyperparameter grids used during model optimization of the harmonisation model and the synthetic NDVI model. During model fitting, a random grid search was conducted with 250 iterations to identify the highest performing set of hyperparameters.**

| Model | Parameter Grid |
|-------|----------------|
| GBM | 'num_leaves': stats.randint(5,50), |
| | 'min_child_samples': stats.randint(10,30), |
| | 'boosting_type': ['gbdt', 'dart'], |
| | 'max_depth': stats.randint(5,25), |
| | 'n_estimators': [300, 400, 500] |

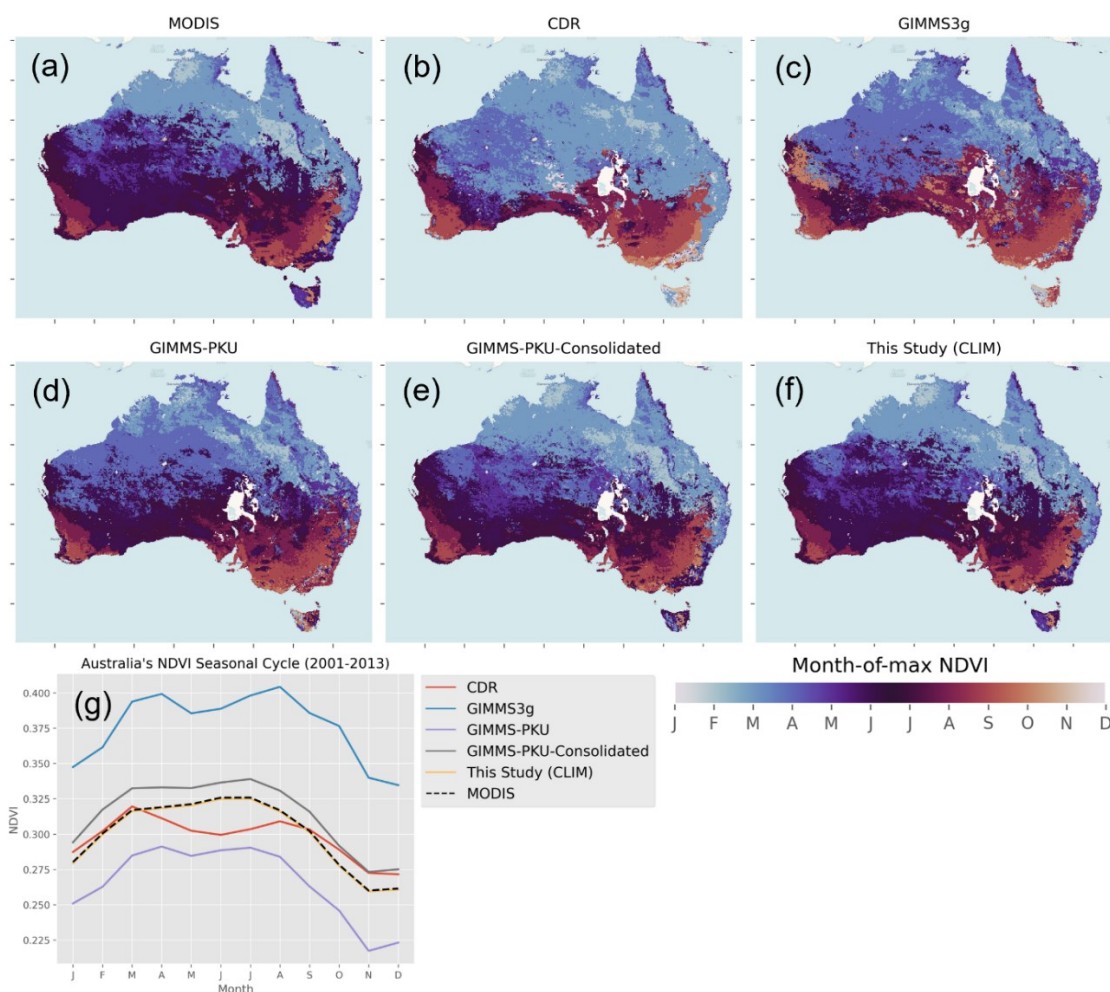

**Figure A3: a-f) Month that maximum NDVI occurs, averaged from 2001-2013, for all NDVI datasets included in the intercomparison between NDVI products, along with the AusENDVI-clim dataset of this study. g) The climatological mean seasonal cycle of NDVI summarised over Australia.**

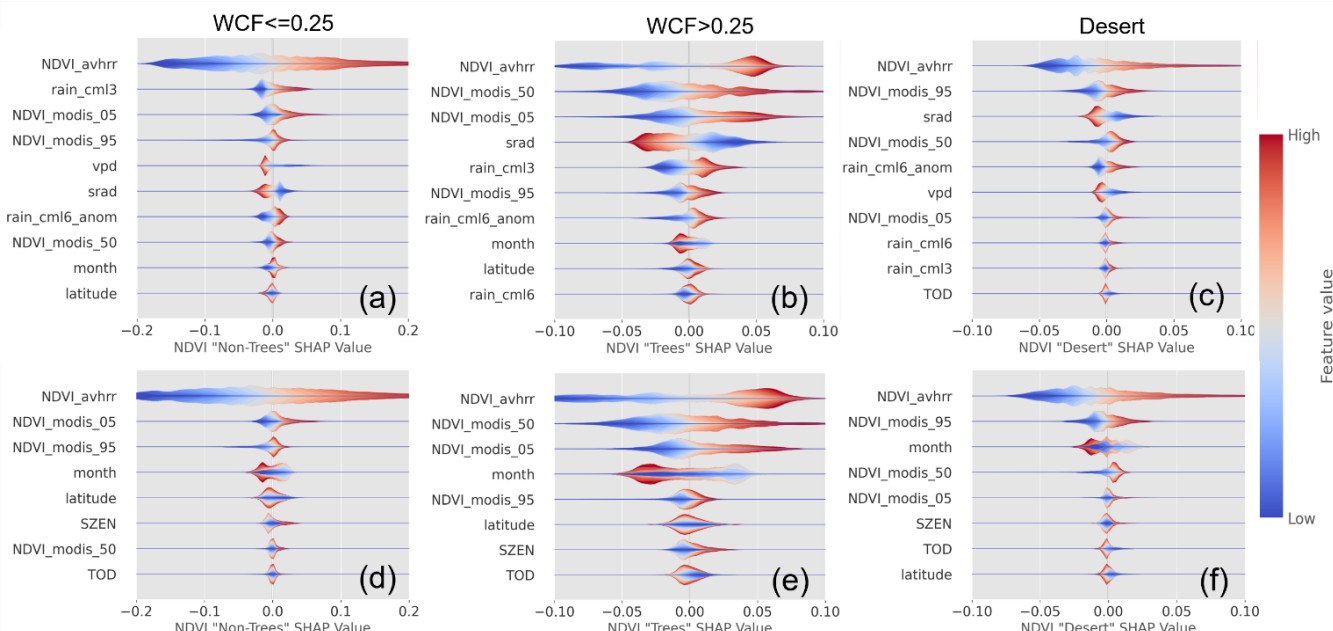

**Figure A4: Feature importance plots for the calibration and harmonisation between NDVI$_{CDR}$ and NDVI$_{MCD43A4}$. a-c) show the results for the 'clim' model. d-f) shows the same but for the 'noclim' model type.**

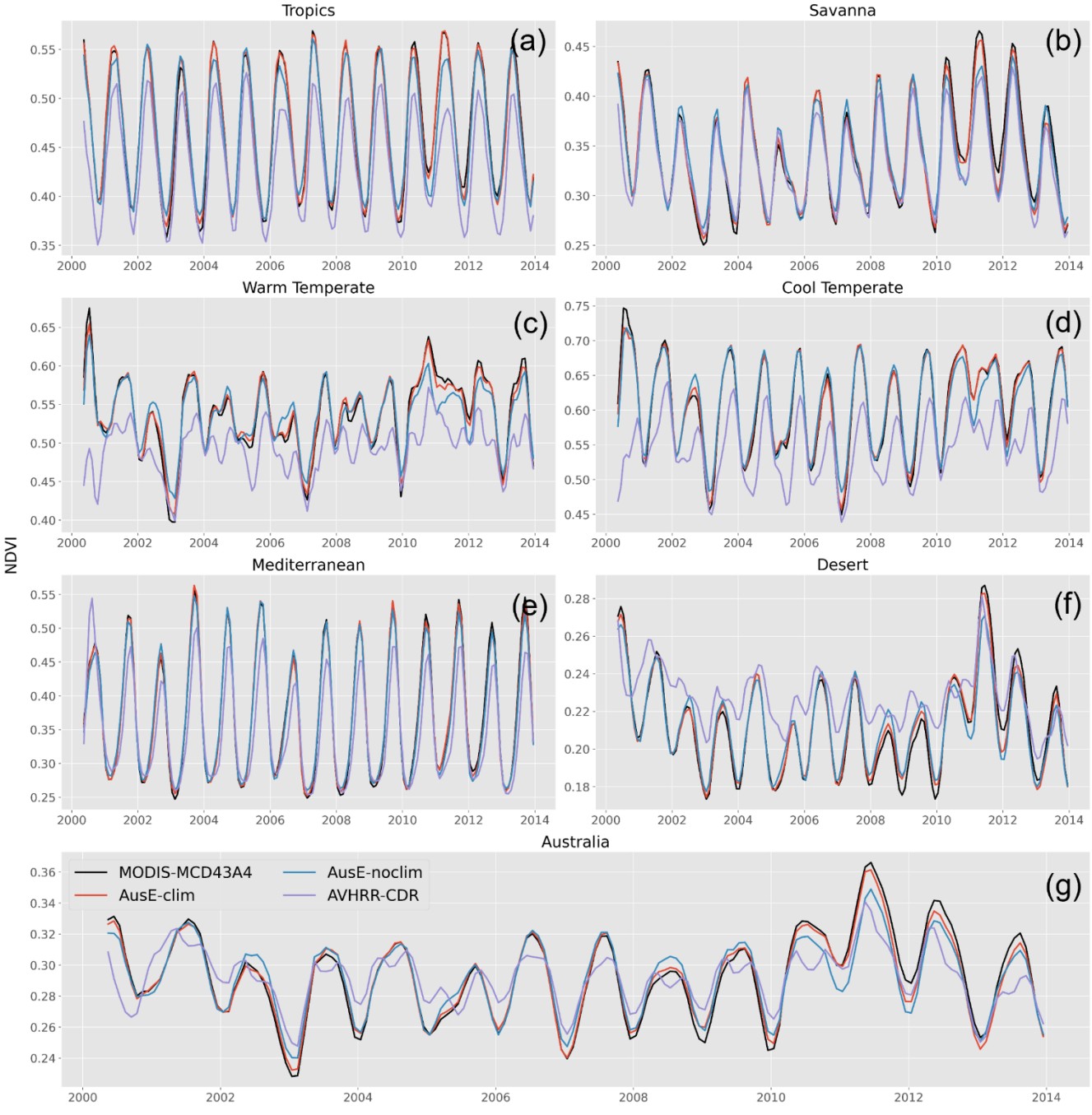

**Figure A5: Per bioregion (a-f) and Australia-wide (g) NDVI time-series before and after the calibration and harmonisation of NDVI$_{CDR}$. Bioregions are defined in Figure 2b. Time series have been smoothed with a three-month rolling mean.**

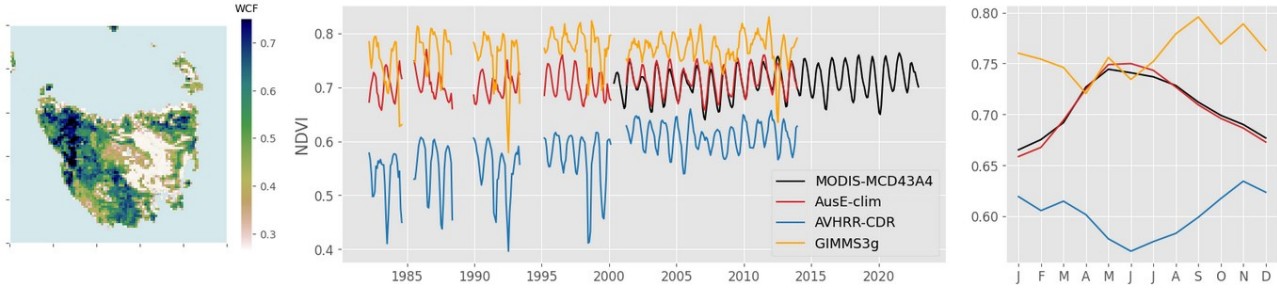

**665**

**Figure A6: Same as Figure 6d-f but including NDVI_GIMMS3g to demonstrate that the very strong increasing trend in NDVI_CDR is likely an artefact of sensor transitions and poor calibration.**

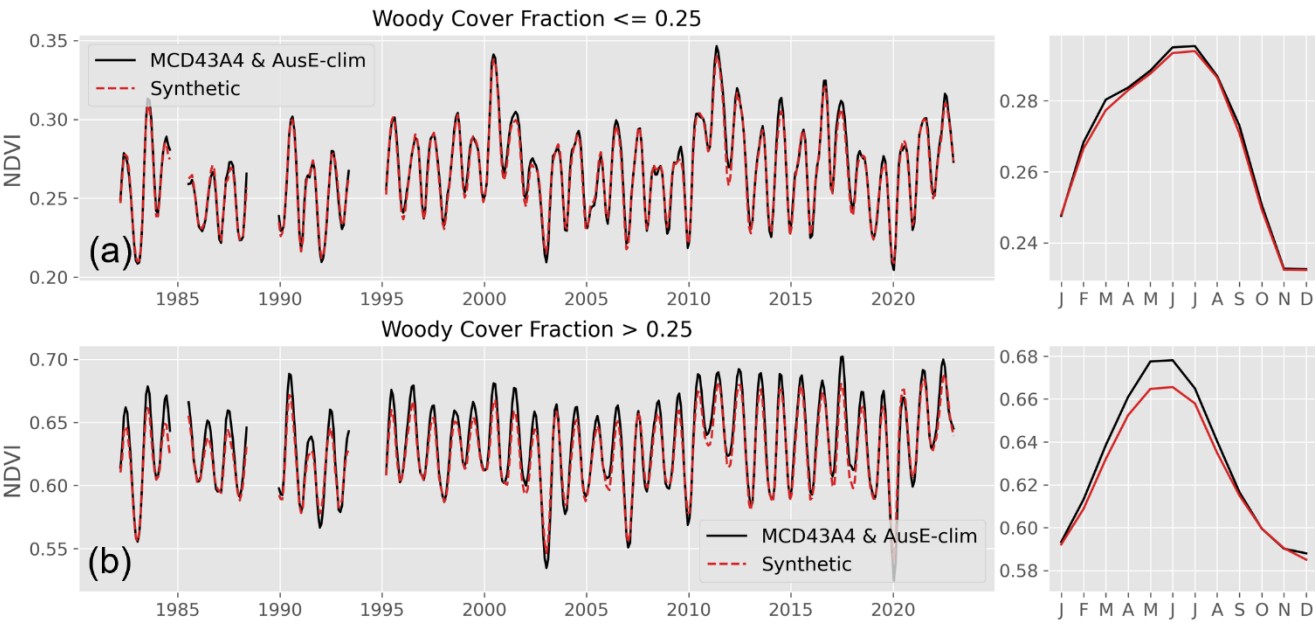

**670**

**Figure A7: Evaluation of the synthetic NDVI built to gap fill NDVI_AusE-clim, disaggregated by high and low WCF regions. a) Spatially averaged observed and synthetic NDVI timeseries over all continental areas where WCF is less than or equal to 0.25. b) Same as (a) but for regions where WCF is greater than 0.25.**

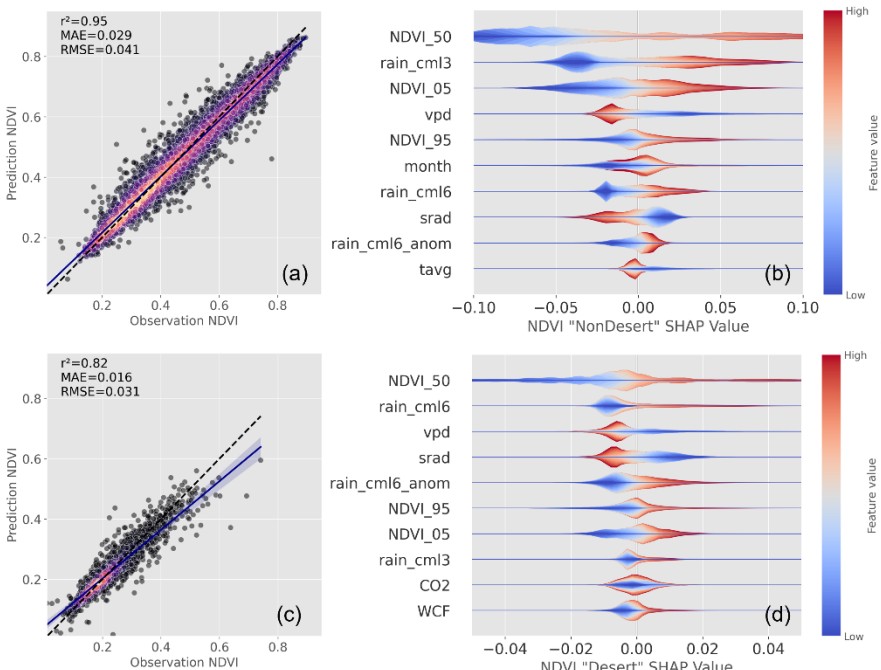

Figure A8: Validation scatter plots and feature importance plots for the gap-filling synthetic NDVI models. a-b) is for the 'nondesert' model region which covers the high and low woody cover regions shown in figure 1a, (c-d) is for the 'desert' region.

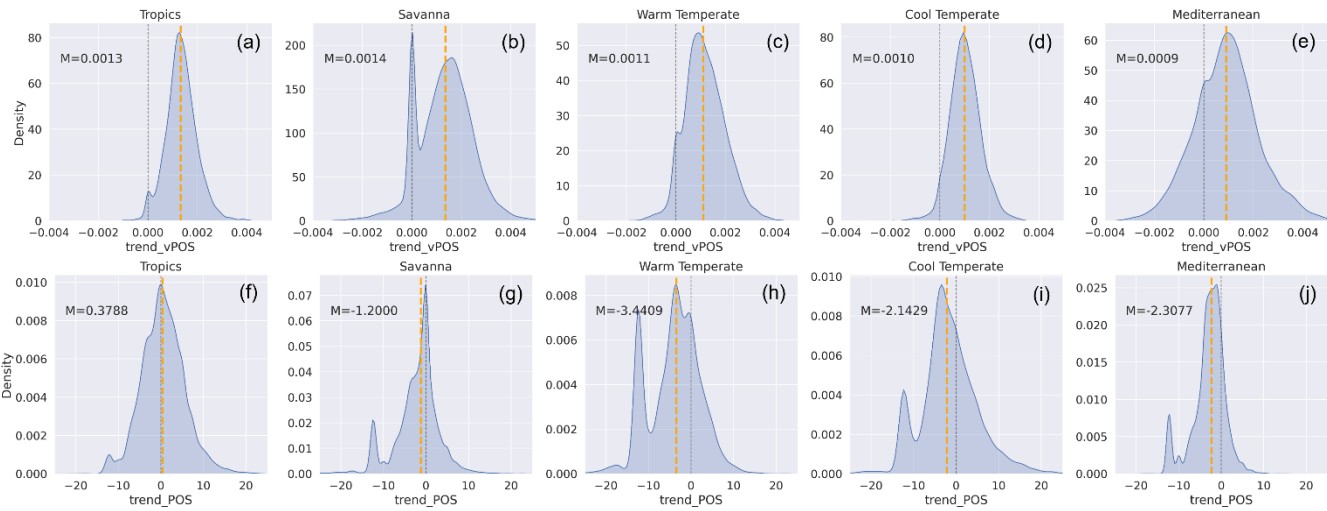

Figure A9: Distributions of pixel level trends in vPOS (a-e) and POS (f-j), summarised by bioclimatic region (excluding the desert region as most of this region is masked as non-seasonal). 'M' refers to the median slope value of the distribution and is indicated by the orange dashed line. Units for vPOS are NDVI per year and units for POS are days per decade.

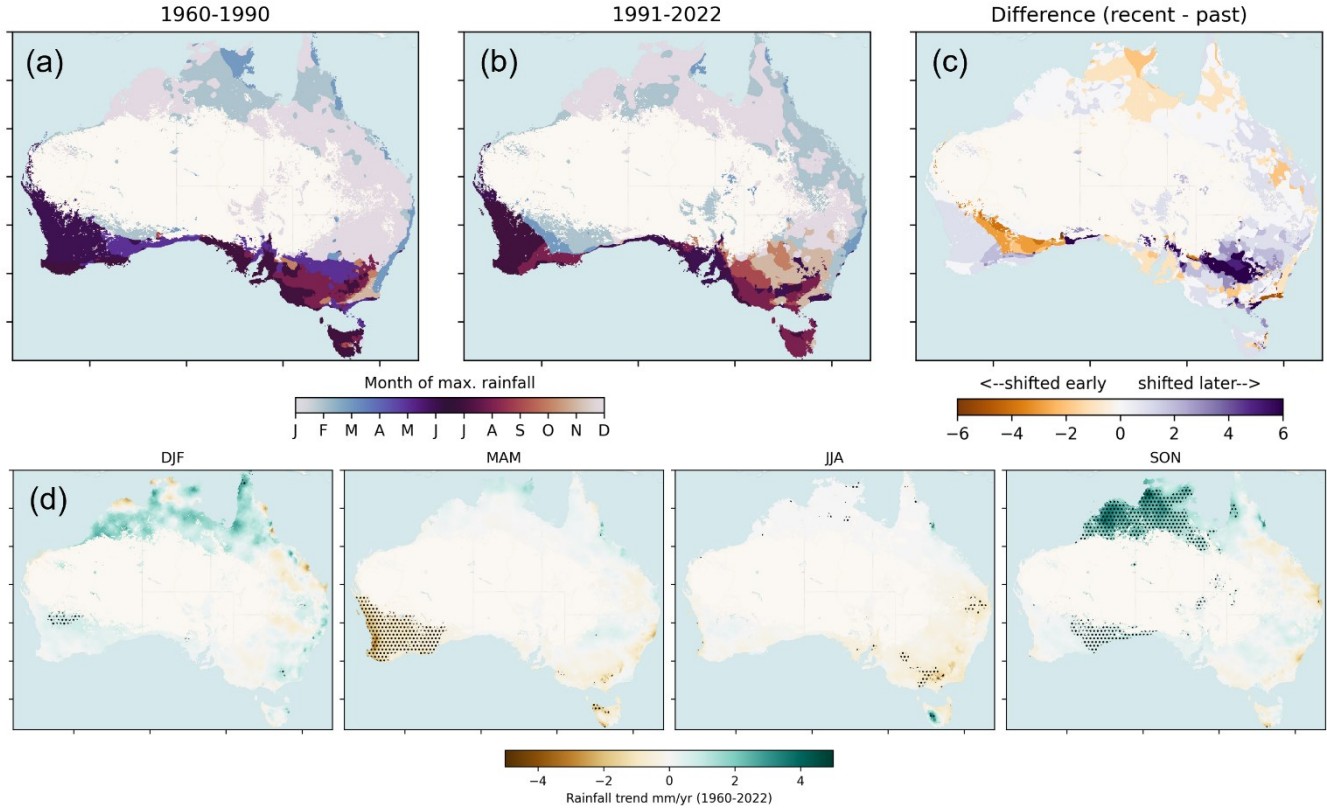

**Figure A10: Changes to the timing and magnitude of rainfall in Australia. a) The typical month that rainfall achieves its maximum value, averaged from 1960-1990. b) Same as (a) but for a 1991-2022 climatology. c) The difference between (a) and (b) where the 1991-2022 climatology is subtracted from 1960-1990. Orange colours indicate earlier peak rainfall in the more recent climatology (in number of months). If peak rainfall shifts from January in 1960-1990 to December in 1991-2022, this is recorded as 'earlier' by one month. Purple colours indicate peak rainfall occurs later in 1991-2022 compared with 1960-1990. If peak rainfall shifts from December in 1960-1990 to January in 1991-2022, this is recorded as 'later' by one month. d) Theil-Sen trends in the total seasonal rainfall from 1960-2022. Hatching indicates significance at 95 % confidence using a Mann-Kendall test.**

## Author Contributions.

CB and SR conceived the study, CB performed all analysis and drafted the manuscript. SR, LR, and AVD provided extensive intellectual input and provided extensive edits to the manuscript.

## Competing interests.

The authors declare that they have no conflicts of interest.

## Acknowledgements

We thank the National Computing Infrastructure (NCI) for providing a research compute environment without which this work would not be possible.

## Financial Support

The first author is supported by a research scholarship provided by Geoscience Australia, funded by the Australian Government.

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
