# Peer review of "Enhancing Long-Term Vegetation Monitoring in Australia: A New Approach for Harmonising and Gap-Filling AVHRR and MODIS NDVI"

_Earth System Science Data, 2024_

## Author Comment (AC1)

**RC1**

**This study proposed several versions of AusENDVI and these NDVIs can be used for studying Australia's changing vegetation dynamics and carbon, and water cycles. The paper is generally organized. The new data set would be useful for the Earth system science studies. However, I still have questions about the structure of the article. Considering these and due to the following major concerns and suggestions, I would recommend it with major revision and to determine whether to accept a revised version.**

**Major concern:**

**RC1-1: I think the article needs a flowchart to show each step, which helps the reader understand the importance of the data processing process. So far I found in the section 'Data and Code Availability' that the author lists each version of AusENDVI, but in fact, I am confused about which step each version of the data is obtained through.**

*We thank the reviewer for noting the need for a flowchart to clarify the methods. We agree that a flowchart would increase understanding, especially since the modelling is broken up into several domains that can get confusing to follow. We had neglected to include one originally in the interests of keeping the number of figures to a minimum but are now convinced, as per your suggestion, that there is a need for it. The flowchart below will be edited into a revised manuscript with the following caption:*

*Figure 1: Flowchart describing the methods of calibration and harmonisation (a), and the development of a synthetic NDVI (b) for gap filling (c). a) Shows the method for the 'clim' model type, the methods for 'noclim' are the same but climate variables are removed from the covariables and 'noclim' is not gap filled. Red coloured boxes denote datasets, blue boxes denote processing steps, and green boxes describe the response variables and covariables used for modelling.*

[Figure]

**RC1-2: I don't think 'Quality of existing NDVIs' is the key part of the article, this part of the results could be replaced by comparing the performance of AusENDVI with other NDVIs, e.g. by adding on the performance of AusENDVI in Figure 2, and then transforming this part into the second part of the results.**

*We respectfully disagree that the section examining the quality of existing NDVI products over Australia is not crucial to the article. We feel it is necessary firstly to help establish the underlying scientific need to develop an Australian-specific long-term NDVI, and secondly to help educate potential users of both AusENDVI and other global NDVI products on the limitations and advantages of these datasets. Of course, there are many ways to structure the article, and your suggestion may increase the overall efficiency of*

*the article by including more results in the same figure set. However, we argue this would reduce the emphasis on the intercomparison and thereby lessen one of the objectives of the study i.e., to determine the suitability of existing NDVI products for long-term vegetation monitoring in Australia.*

**RC1-3: I think the first part of the results could be to highlight the results of each step, especially 'before and after the calibration and harmonization' and 'before and after gaping fill'. Of course, these are already in the results, but they should be in the same section to highlight the results of each step of the enhancement.**

*We believe it is necessary for the results of the calibration and gap filling to be in different subsections as there are two distinct modelling efforts occurring here. In the first instance (section 3.2), we report the results of harmonising AVHRR to MODIS, and in section 3.3 we report the results from the creation of a synthetic NDVI for gap filling. The creation of a synthetic NDVI is a different enough process from the harmonisation that we argue it requires its own subsection (different models, modelling domains, and input data). Note also that the two sub-sections immediately follow each other so narratively the current structure of the article is similar to your suggestion. We hope the inclusion of a flowchart will also help clarify this and make more obvious the need to separate the results of the two different steps.*

*We also aim to include the plot below into Figure 7 to show the 'before and after' results of gap-filling. As missing data tends to be in the higher NDVI regions (wet, cloudy, forested regions), gap filling has the tendency of increasing NDVI when averaging over the continent. Figure A4 in the current manuscript also shows the time series of CDR-AVHRR before and after the calibration/harmonisation, averaged across all of Australia and broken down by bioclimatic region. We are open to including this in the main part of the manuscript at the editor's discretion.*

[Figure]

**RC1-4: Is it possible to find field measurements of NDVIs in Australia to provide absolute accuracies for individual NDVIs, and if so, this would be an important support for demonstrating the accuracy of AusENDVIs.**

*In short, no. There are no in-situ field measurements of NDVI that are comparable to the spatial and temporal scales of AusENDVI (the area of pixels in AVHRR are ~25 km$^2$). However, note that MODIS MCD43A4 surface reflectance data (from which we calculate NDVI as the response variable for the harmonisation) is a well calibrated and validated remote sensing product, and the validation performed in our study is based on random pixels selected from MODIS. We also included a comparison with the Digital Earth*

*Australia Landsat surface reflectance product as this product has all the same types of corrections (atmospheric, BRDF etc.) (Byne et al. 2024) as MODIS MCD43A4 and is therefore a fair and independent inter-comparison dataset.*

**RC1-5: The discussion is too lengthy, my suggestion is that it could be broken up into subsections.**

*We will revise the manuscript to include subtitled sections in the Discussion, and where possible we will edit for clarity and brevity.*

**RC1-6: As with 'Trends in peak-of-season phenology', I would suggest that the authors do the same study again, using the available NDVIs, do a trend analysis of annual averages, and then compare it to the results in the literature, to sidely bolster the credibility of these data.**

*We appreciate the reviewer's suggestion here and below we have assessed the annual-average NDVI trends across Australia for the different NDVI products to see how they differ. AusENDV-Clim closely reproduces the observable trends in MODIS and GIMMS3g (coefficients: AusECLIM=0.00058 NDVI yr⁻¹, MODIS=0.00066 NDVI yr⁻¹, GIMMS3g=0.00061 NDVI yr⁻¹). Trends in the two GIMMS-PKU products are less than half those of MODIS and GIMMS3g. This result reinforces our previous assertions that no pre-existing AVHRR-based NDVI product both reproduces close agreement with the MODIS record while simultaneously reproducing satisfactory results in the pre-MODIS era. We aim to include this analysis in a revised manuscript.*

[Figure]

*Figure: Annual average NDVI trends summarised over Australia for the overlapping period of 1982-2013. All data gaps have been matched between datasets, and datasets have been reprojected to match the resolution of GIMMS3g. Trend lines have been fitted using ordinary least-squares regression and coefficients are expressed in terms of NDVI yr⁻¹.*

*References*

*Byrne, G., Broomhall, M., Walsh, A. J., Thankappan, M., Hay, E., Li, F., ... & Denham, R. (2024). Validating Digital Earth Australia NBART for the Landsat 9 Underfly of Landsat 8. Remote Sensing, 16(7), 1233.*

---

## Author Comment (AC2)

**RC2**

In the manuscript titled "Enhancing Long-Term Vegetation Monitoring in Australia: A New Approach for Harmonising and Gap-Filling AVHRR and MODIS NDVI", Burton et al. reconstructed new harmonised NDVI datasets in Australia using the GBM method. The manuscript and figures are well prepared. I appreciate the extensive work conducted in this study, like, comparing existing datasets, producing new datasets and applications. However, from my perspective, this paper may still lack sufficient novelty to warrant publication in ESSD. Below, I outline my main concerns and provide point-to-point comments.

**Main concerns:**

**RC2-1:** In the context of the existing abundance of NDVI datasets such as VIP15 NDVI, GIMMS NDVI3g and the latest PKU NDVI, authors still aim to produce new NDVI datasets, which is challenged. I encourage this work, but authors fail to show strong motivations for doing so (like, data unavailability or any issues present in existing datasets).

*We wholeheartedly agree that there is an abundance of existing global NDVI datasets, and we have gone to considerable effort to include many of the most prominent datasets in a detailed intercomparison. In the introduction, we list several well-known discrepancies with existing NDVI products (lines 66-70), and also make note that the recent PKU-GIMMS product has yet to be widely assessed by the community owing to its recent release. This is why we set our first objective of the study to assess many of the pre-existing datasets to determine if they are suitable for studying the long term biogeophysical impacts of global change on Australia's terrestrial vegetation. Note that while there are many studies at the global scale that assess existing NDVI products, none have focused on Australia, and we see this inter-comparison as itself a valuable contribution to the Australian research community.*

*Ultimately, we conclude that GIMMS3g, CDR, and GIMMS-PKU have significant deficiencies (sensor transition issues, poor correlation, and/or high error with MODIS). GIMMS-PKU-consolidated offers a real improvement over other products, however, GIMMS-PKU-consolidated still has shortcomings, primarily that it does not display realistic inter-annual variability in the 1982-2001 period, and displays a lower trend in annual average NDVI from 1982-2013 than GIMMS3g and AusENDVI (figure and comments on annual average trends are in the next response, RC2-2). Hence, we argue there are further advances that can be made by optimising to the regional scale, by including a range of new features such as climate variables in the calibration, and by developing a more robust gap-filling technique. In short, our aim is to develop the best possible NDVI dataset optimised for the needs of the Australian research community, that iteratively improves on previous datasets, just as GIMMS-PKU iteratively improved on GIMMS3g.*

**RC2-2:** According to the results (like, figures 2 & 8), I think PKU-consolidated dataset has been produced well, and compared to PKU data, your dataset does not show any significant and necessary improvements. Therefore, I would suggest highlighting clear improvements than other existing datasets.

The recent release of the GIMMS-PKU-consolidated dataset showed significant improvements over previously existing global NDVI datasets as it effectively remediated some sensor transition issues, aligns well with MODIS, and, at the global scale, better reproduced the greening trend observable in MODIS. However, over Australia, it is our contention that it fails to reproduce realistic inter-annual variability in the pre-MODIS era as indicated by its lack of agreement with the Landsat record in Figure 3a, and the distinct lack of rainfall-driven inter-annual variability as shown in Figure 3b and Figure 8b, respectively. This is important as the terrestrial biosphere's response to climate extremes (droughts, heavy rainfall) is of paramount importance to study given the changing frequency of climate extremes in Australia (Lewis et al. 2017). How Australia's ecosystems are responding to these changes may depend on the shifting seasonality of rainfall, warming air temperatures, and increasing atmospheric $CO_2$ concentrations which all affect plant physiology. We cannot effectively study these impacts and mechanisms (at the continental scale) if vegetation variability from 1982-2000 is artificially subdued.

In the figure below we develop the statistical relationships between twelve-month rolling mean standardised rainfall and NDVI anomalies, averaged across Australia for different periods and different products. If we consider the slope of the linear relationship between rainfall and NDVI to be a reasonable approximation of the sensitivity of NDVI to water supply (and we assume there should be approximate stationarity in these relationships), then AusENDVI-clim in the 1982-2000 period (c) displays a similar sensitivity and correlation as MODIS does in the 2000-2022 period (b). Contrast this with GIMMS-PKU-consolidated which has a substantially lower sensitivity in the 1982-2000 period (d) than it does in the 2000-2022 period (e) (approximately half the sensitivity). While we may expect some changes in water-supply sensitivity over the decades due to effects such as $CO_2$ fertilisation, a doubling of water-supply sensitivity is highly unlikely. It is clear that AusENDVI is responding more realistically to rainfall-driven interannual variability than GIMMS-PKU-consolidated, which we consider an iterative advancement. We will include these scatter plots in an updated manuscript, along with the time series of AusENDVI-clim and GIMMS-PKU-consolidated anomalies (i.e., we will update figure 8 with these plots and adjust the results/discussion accordingly).

[Figure]

Figure: a) Standardised NDVI anomalies of AusENDVI-clim (1982-2000) merged with MODIS MCD43A4 (2000-2022), and GIMMS-PKU-consolidated. Both datasets have been gap-filled identically following the methods described in section 2.3.   b-d) Relationships between twelve-month

*rolling mean standardised rainfall and NDVI anomalies averaged across Australia for different periods. Rainfall, AusENDVI and GIMMS-PKU-consolidated anomalies have been calculated against a 1982-2022 baseline. MODIS NDVI anomalies have been calculated against a 2000-2022 baseline. The relationship y=mx+c denotes the linear regression slope between rainfall and NDVI anomalies where y is NDVI anomalies, x is rainfall anomalies, and m is the slope coefficient. The slope coefficient can be considered an approximation of the sensitivity of NDVI to anomalous water supply.*

*Additionally, in the second figure below we show the annual average NDVI trends across Australia for the assessed NDVI products. Trends in the two GIMMS-PKU products are less than half those of MODIS, GIMMS3g, and AusENDVI. This result reinforces our assertion that no pre-existing AVHRR-based NDVI product can both reproduce close agreement with the MODIS record while simultaneously reproducing satisfactory results in the pre-MODIS era. We aim to include the annual average trend analysis in a revised manuscript.*

[Figure]

*Figure: Annual average NDVI trends summarised over Australia for the overlapping period of 1982-2013. All data gaps have been matched between datasets and datasets have been reprojected to match the resolution of GIMMS3g. Trend lines have been fitted using ordinary least-squares regression and coefficients are expressed in terms of NDVI yr⁻¹.*

*To summarise, the advantages of AusENDVI are that: 1) it closely reproduces the MODIS record in terms of seasonality, interannual variability, and trends in annual-average NDVI, 2) it reproduces anomalies in the Landsat NDVI record in the pre-MODIS era (back to 1988), and shows realistic rainfall-driven interannual variability back to 1982, 3) gap-filling in AusENDVI does not rely on methods such as filling with a climatology, spatial interpolation methods, or lengthy temporal interpolation methods that are unreliable where wide-spread and lengthy data-gaps occur, 4) it has a higher spatial resolution than any of the GIMMS datasets and is built using inputs that apply the full suite of atmospheric and BRDF corrections, and 5) the methods and code for its development are entirely open-sourced. No other existing product can lay claim to all these attributes which is why we argue AusENDVI is a worthwhile addition to the suite of NDVI products available.*

**Other comments:**

**RC2-3: No ground observations (like, Flux or PhenoCam sites) to validate your data?**

*It is unlikely that eddy-covariance flux tower GPP would have a proportional relationship with NDVI at the 5 km scale, and across the many different land covers (Camps-Valls et al 2021). Likewise, the small phenocam network in Australia does not record NDVI values. Instead, they record RGB images that can be converted to 'green chromatic coordinate' values but GCC values are not directly comparable to NDVI (Hufkens et al. 2018, St Peter et al. 2018). Regardless, there still exists a large mismatch in spatial and temporal scales between phenocams and AusENDVI (or any other AVHRR NDVI dataset, the area of pixels in CDR-AVHRR are ~25 km$^2$). Hence, there is no ground validation data for an independent assessment of our data. However, note that MODIS MCD43A4 surface reflectance data (from which we calculate NDVI as the response variable for the harmonisation) is a well-calibrated and validated remote sensing product, and the validation performed in our study is based on random pixels selected from MODIS. Likewise, we also include a comparison with the Digital Earth Australia Landsat surface reflectance product as this product has all of the same types of corrections (atmospheric, BRDF etc.) (Byne et al. 2024) as MODIS MCD43A4 and is therefore a fair and independent inter-comparison dataset.*

**RC2-4: For any designed steps (e.g., gap filling), it is expected to see the comparison of results for before and after processing (can refer to the guide: https://lpdaac.usgs.gov/documents/1328/VIP_User_Guide_ATBD_V4.pdf).**

*For the gap-filling, we will insert the figure shown in our response to RC1-3. Figure A4 in the current manuscript shows the time-series of CDR-AVHRR before and after the calibration/harmonisation, averaged across all of Australia and broken down by bioclimatic region. We are open to including this in the main part of the manuscript at the editor's discretion.*

**RC2-4: Add a flowchart to summarize each step and processing.**

*We thank the reviewer for this suggestion and we will include in the revised manuscript the flow-chart shown in our response to RC1-1.*

**RC2-5: Add some quantified results in the abstract to show the reliability/enhancement of your datasets.**

*We will add the statistics from Figure 4 to the abstract to show the model agreements with observation, along with the statistics from Figure A5 that shows the agreement between the synthetic NDVI and observations.*

**RC2-6: Lines 30-35, provide spatial and temporal resolutions information for your 41-year dataset.**

*We will include the spatial and temporal resolution in the abstract in a revised manuscript.*

*References*

*Lewis, S. C., Karoly, D. J., King, A. D., Perkins, S. E., & Donat, M. G. (2017). Mechanisms explaining recent changes in Australian climate extremes. Climate Extremes: Patterns and Mechanisms, 249-263.*

*Camps‑Valls, G., Campos-Taberner, M., Moreno-Martínez, Á., Walther, S., Duveiller, G., Cescatti, A., ... & Running, S. W. (2021). A unified vegetation index for quantifying the terrestrial biosphere. Science Advances, 7(9), eabc7447.*

*Hufkens, K., Filippa, G., Cremonese, E., Migliavacca, M., D'Odorico, P., Peichl, M., ... & Wingate, L. (2018). Assimilating phenology datasets automatically across ICOS ecosystem stations. International agrophysics/International Advertising Association.-New york, 32(4), 677-687.*

*St. Peter, J., Hogland, J., Hebblewhite, M., Hurley, M. A., Hupp, N., & Proffitt, K. (2018). Linking phenological indices from digital cameras in Idaho and Montana to MODIS NDVI. Remote Sensing, 10(10), 1612.*

*Byrne, G., Broomhall, M., Walsh, A. J., Thankappan, M., Hay, E., Li, F., ... & Denham, R. (2024). Validating Digital Earth Australia NBART for the Landsat 9 Underfly of Landsat 8. Remote Sensing, 16(7), 1233.*

---

## Author Comment (AC3)

**RC3**

**This manuscript by Burton et al. proposes a new long-term NDVI dataset specifically for Australia (AusENDVI) by harmonizing and gap-filling AVHRR and MODIS data. Compared to global NDVI datasets, localized AusENDVI could provide optimized NDVI observation with the aid of prior knowledge. To this end, I agree that the AusENDVI could be a promising dataset for better understanding long-term vegetation dynamics in Australia. However, the current manuscript faces many major issues and lacks essential information that shows the superiority of AusENDVI. My overall attitude is somewhere between a severely major revision and rejection. That's dependent on how the authors respond to the following comments.**

**Major comments:**

**RC3-1:** First, NDVI is a spectral index calculated from red and near-infrared reflectance. Discrepancies of band settings (spectral range, FWHM, etc.) between sensors could be an important driver of the NDVI difference. This is the case for the three types of sensors involved in the manuscript, i.e., Landsat TM/ETM+, AVHRR, and MODIS. However, this source of NDVI differences in band setting has been completely ignored in the evaluation of current global NDVI datasets and generation of AusENDVI. For example, the authors failed to compare the two reference datasets, Landsat TM/ETM+ and MODIS in the manuscript.

*We wholly agree with the reviewer that spectral sampling is different between sensor specific time-series, and we ought to have raised this point in the introduction and discussion sections, which we will do in an updated manuscript.*

*However, the main reason for calibrating and harmonising AVHRR to MODIS is largely to ameliorate the differences in spectral sampling between the sensors so they can be combined to produce a consistent time-series. Spectral sampling differences also cannot explain the fairly large inconsistencies between AVHRR-based products such as CDR, GIMMS3g, and GIMMS-PKU, for example in the seasonal cycles shown in Figure 2e. These differences must be due to other aspects of their processing such as those we listed in the discussion (different atmospheric corrections, cloud contamination, gap-filling procedures, etc.).*

*We thank the reviewer for pointing out that we had neglected to include a comparison between MODIS and Landsat. We will include the time-series pictured below of Landsat vs MODIS anomalies in the appendix of an updated manuscript. As you can see, in terms of inter-annual variability, there is very good agreement between the sensors. The differences in spectral sampling between MODIS and Landsat are why we use Landsat only for validating inter-annual variability in the pre-MODIS era (using annual anomalies aggregated over the continent), since mean differences in NDVI between sensors are removed by anomalies.*

[Figure]

*Figure: Standardised anomalies of the overlapping period between MODIS MCD43A4 NDVI and Landsat NDVI derived from the common baseline period of 2000-2012. Rainfall anomalies are derived from a longer baseline of 1982-2022.*

**RC3-2:** **Second, for some reason, the temporal resolution of the AusENDVI has been missing in the Abstract and Conclusion section of the manuscript. For a long-term dataset, the temporal resolution is a critical attribute that determines how well the AusENDVI could capture the abrupt vegetation changes due to climate or anthropogenic disturbances. As far as I could find in the manuscript and the data repository, AusENDVI provides monthly data records. It could be disappointing because the temporal resolution of current global NDVI datasets such as NDVI3g and NDVIpku is half a month. This issue is related to another one in that AusENDVI uses median composites while NDVI3g, NDVIpku, and MODIS NDVI use maximum composites. Why is the median? Will that underestimate vegetation growth such as vPOS?**

*We apologise for not making explicit in the abstract and conclusion that it is a monthly product, we will highlight the temporal resolution in an updated manuscript.*

*There are quite a few advantages to aggregating to the monthly scale. Firstly, it reduces the concerns of matching overlap times between all the differently sourced datasets and thereby increases comparability between datasets. Secondly, and most importantly, it helps lessen the impact of noisy sub-monthly signals that arise from unmasked residual clouds etc. that imperfect QC bands miss. And lastly, it makes deriving relationships between covariables like climate simpler as a number of these variables come as monthly aggregates. Moreover, we argue that a monthly product is sufficient for all the likely use cases of this dataset: monitoring long-term changes to vegetation due to global environmental change ($CO_2$, warming, rainfall changes), and for use in driving or validating land surface models. AusENDVI has a coarser temporal resolution than the GIMMS products, true, but it also has the not-insignificant advantage of a higher spatial resolution.*

*Both maximum and median compositing techniques are robust to outliers, which is the principal reason for using them. It's true that GIMMS3g and GIMMS-PKU use maximum compositing techniques in their development of a 16-day product, but when we loaded these datasets (and the MODIS 16-day product) we took monthly medians so the differences between reanalysing the data using max instead of median is likely to be small (i.e. the difference between the median of two values in a month or the highest of two values in a month), and differences should only occur in the overall mean NDVI*

*response not temporal dynamics. However, to test this, we would be quite willing to rerun the analysis with maximum-value compositing if so requested.*

*There is no reason to expect that trend values in vPOS will change substantially by switching from median to maximum compositing, though the actual values of vPOS may increase marginally. We argue such a change would not be of material consequence.*

**RC3-3:** **Third, the most impressive feature of AusENDVI is that it accounts for the dominant role of precipitation in Australia. However, the strong relationship between precipitation and NDVI has been an unproved precondition in the manuscript. The authors must demonstrate pixel-wise precipitation-NDVI relation before the relationship is used to evaluate NDVI products and generate AusENDVI. For example, in Figure 8b, the abrupt increase of NDVI in 1984 does not seem to follow the precipitation anomalies (Note the authors use the precipitation anomalies to argue the deficiency of other NDVI products). A literature review without a pixel-wise relation map is not enough.**

*We apologise for not including any figures demonstrating the strongly water-limited nature of vegetation in most of Australia - this is an oversight that stems from our familiarity with the landscape. We can include in the appendix the below per-pixel maps of the correlation between NDVI anomalies and precipitation anomalies to demonstrate this relationship. The figure highlights that cumulative rainfall anomalies are strongly correlated with NDVI anomalies over the majority of Australia's land mass.*

[Figure]

*Figure: Pearson correlations between NDVI anomalies and monthly, three-monthly, and six-monthly cumulative rainfall anomalies.*

*The apparent large peak in NDVI in 1983-1984 in figure 8b is accentuated by being bracketed by a severe drought in 1982, and modest droughts in 1985-86 (both 1983 and 84 were above-average rainfall years though - interactive maps of annual rainfall anomalies can be examined here). When shown as a standardised anomaly against a 40-year baseline it is in fact a fairly modest positive anomaly (see figures in RC2-2). We will update Figure 8 to include the figure shown in our response to RC2-2 so anomalies are easier to deduce. Furthermore, although the rainfall stripes are a nice visualisation of aggregate rainfall patterns, they conceal much about the seasonal timing and spatial allocations of rainfall within a given year that can matter for the vegetation response (again, cumulative effects are also very important). In the figure shown in our response to RC2-2, we also include the statistical relationships between twelve-month rolling mean standardised rainfall and NDVI anomalies, averaged across Australia for different periods and different products. If we consider the slope of the linear relationship between rainfall and NDVI to be a reasonable approximation of the sensitivity of NDVI to water supply (and we assume there should be approximate stationarity in these relationships), then AusENDVI-clim in the 1982-2000 period (c) displays a similar*

*sensitivity and correlation as MODIS does in the 2000-2022 period (b). Contrast this with GIMMS-PKU-consolidated which has a substantially lower sensitivity in the 1982-2000 period (d) than it does in the 2000-2022 period (e) (approximately half the sensitivity). While we may expect some changes in water-supply sensitivity over the decades due to effects such as $CO_2$ fertilisation, a doubling of water-supply sensitivity is highly unlikely. It is clear that AusENDVI is responding more realistically to rainfall-driven interannual variability than GIMMS-PKU-consolidated, which we consider an iterative advancement.*

**RC3-4: Last, the authors failed to demonstrate the improvements of AusENDVI in critical aspects such as long-term trends of vegetation and SOS.**

*In terms of long-term trends, please refer to the comments and figures made in our response to RC1-6.*

*On the trends in phenology (we assume the reviewer meant 'POS'), it would be our preference not to include another large figure and discussion intercomparing phenology trends between datasets. The phenology analysis was included as a short use case of AusENDVI to demonstrate its capability, but otherwise, the paper is intended to focus on the derivation of the data. We have plans to do a broader analysis of phenology trends (and examine a broader range of phenometrics) in subsequent work and it is our contention that including such an analysis here is unnecessary as we already demonstrate the merit of AusENDVI through a number of figures. Also, note that figure A2 does show some of the differences between products for the average month-of-maximum NDVI (averaged over the years 2000-2013).*

*To summarise, the advantages of AusENDVI are that: 1) it closely reproduces the MODIS record in terms of seasonality, interannual variability, and trends in annual-average NDVI, 2) it reproduces anomalies in the Landsat NDVI record in the pre-MODIS era (back to 1988), and appears to show realistic rainfall-driven interannual variability back to 1982, 3) gap-filling in AusENDVI does not rely on methods such as filling with a climatology, spatial interpolation methods, or lengthy temporal interpolation methods that are unreliable where wide-spread and lengthy data-gaps occur, 4) it has a higher spatial resolution than any of the GIMMS datasets and is built using inputs that apply the full suite of atmospheric and BRDF corrections, and 5) the methods and code for its development are entirely open-sourced. No other existing product can lay claim to all of these attributes which is why we argue AusENDVI is a worthwhile addition to the suite of NDVI products available.*

**Some minor but still important comments:**

**RC3-5: Line 96-97. Why are SOS and EOS not included?**

*As per our last point, we have plans to do a broader analysis of phenology trends, and examine a broader range of phenometrics and their drivers in subsequent work and don't wish to overload a dataset-description paper with too much 'applications' content.*

**RC3-6: Line 104. When is averaging used and when is nearest-neighboring used?**

*Averaging is used when a finer-resolution dataset is downsampled to coarser resolution (e.g. 5km → 8km), and nearest-neighbour sampling is used when two datasets have the same/similar spatial resolution but either different projections or slightly different grid extents. We will include this information in an updated manuscript.*

**RC3-7: Line 105. How to deal with the radiometric difference between Landsat TM and ETM+ (Berner et al., 2020; https://doi.org/10.1038/s41467-020-18479-5)?**

*Firstly, the specific Landsat datasets we used, Digital Earth Australia's surface reflectance NBAR product - Collection 3, is a very high quality and consistent surface reflectance product that is calibrated and validated to the Australian continent (Byrne et al. 2024). These corrections minimise the differences between sensors (the paper you linked uses USGS C1 Landsat which is less processed/corrected than DEA-NBAR). To demonstrate this, the figure below summarises NDVI over Australia for the year 2004 (the first year when both sensors are running for the full length of the year). These time series broadly match in terms of variability and magnitude, though LS7 has a slightly higher mean NDVI. Note that when acquiring the Landsat data, all LS5 and LS7 data are averaged together for a given month, so the actual values used in the manuscript sit between the lines (purple line in the plot below).  Secondly, for the years 1988-2000 the time series only consists of Landsat 5 which is the key period we are assessing. Lastly, we only use Landsat for assessing annual anomalies so any differences in mean responses between sensors are minimised.*

[Figure]

**RC3-8: Line 212. Why is the median rather than the maximum value?**

*We are unsure what the reviewer refers to here as line 212 does not discuss medians, but in general, we use medians for temporal compositing to reduce the influence of outlier values.*

**RC3-9: Line 122. Please provide more information on the use of the quality assurance band.**

*We will include more specifics in the text, but the notebooks in the linked github repository also detail how data was loaded and masked (see here).*

**RC3-10: Line 128. Simply removing data in sensor transition would not only eliminate the gradual effect of sensor degradation but also the valuable information of NDVI anomaly. Note the eruption of Pinatubo (1991) and the transition of AVHRR2 and AVHRR3 (around 2000) are not accounted for.**

*We remove data at the sensor transitions of the CDR product because the large anomalies associated with some of these transitions are unrealistic and do not reflect conditions on the ground. For example, see Figure 2 in Tian et al. (2015), where*

*transitions between AVHRR sensors N9 and N11 result in an extremely large negative anomaly.*

*We do not remove data during or immediately after the Pinatubo eruptions in 1991. We do remove data associated with the transition from sensors N11 to N14 during the second half of 1993 through 1994, though there is little-to-no data recorded in the CDR product over Australia in these years anyway. It is our understanding that after ~3 years the impact of aerosols from Mt Pinatubo on the NDVI signal waned. It is important to note that the CDR product includes an aerosol correction, something the GIMMS products generally lack; we understand that GIMMS3g does include a correction specifically for the Pinatubo eruption but does not have aerosol corrections otherwise.*

*We did not remove data at the year 2000 transition from sensors N14 to N16, although a lot of data is missing in this period so its influence on the time-series is probably limited.*

**RC3-11: Line 131 & Figure A1. Explain the reason why some regions experience lower data availability. How does the data availability affect the evaluation of NDVI products and AusENDVI accuracies?**

*Coastal, alpine, and tropical regions experience fewer 'good quality' observations principally due to the greater abundance of clouds in these regions from prevailing weather systems. For example, western Tasmania experiences >1500 mm of rainfall per year which means the chances of acquiring a clear satellite image are lower. A map of annual mean rainfall over Australia can be viewed here. The patterns of high rainfall largely coincide with regions of lower data quality. Exceptions to this 'rule' sometimes occur where bright objects (e.g. salt lakes) can be misidentified as cloud by quality-assurance bands, such as is the case in parts of central arid Australia.*

*Whenever products are compared in the manuscript, all datasets are reprojected onto a common grid and data gaps are matched between all datasets. Basically, a mask is created that identifies all missing pixels in all datasets, and then that common mask is applied to every dataset. This ensures a fair and valid comparison. Even in those areas with a lower data volume, there are still dozens of valid pixels in the time series so statistics are fairly robust. One caveat to this might be that, in those very cloudy regions (e.g. tropical forests along northern Queensland), the statistics probably relate the agreement mostly during the dry seasons as the wet seasons will have fewer observations.*

*It is also reiterated that the differing volumes of data across the continent are partly why we implemented a stratified, equalised random sampling approach for the training and validation samples. Providing the same number of samples for each bioclimatic region regardless of data availability or area reduces bias in validation statistics.*

**RC3-12: Table 1. Please provide the temporal resolution of the datasets.**

*We will include the 'native' temporal resolutions in an updated version of the manuscript.*

**RC3-13: Line 137. Why not use existing MODIS NDVI products (MOD13Q1, MOD13C1, etc.)? It looks like AusENDVI and NDVIpku are based on different MODIS products. Will be the difference reflected in the evaluation of NDVIpku?**

*We used MODIS MCD43A4 as we argue it is the highest quality MODIS dataset that been released owing to: 1) its full suite of atmospheric corrections, 2) BRDF corrections,*

and 3) inclusion of both the AQUA and TERRA satellites which extends its time-range back to the year 2000.  Also, the inclusion of BRDF corrections aligns MODIS with both the AVHRR-CDR product and the Landsat product.  It is true that the GIMMS-PKU-consolidated product was trained on a different version of MODIS, but we make no contentions in the paper that GIMMS-PKU-consolidated does not align well with MODIS - it does. Presumably comparing it with the exact version it was trained on would show further agreement, but given we ourselves did not wish to use that same version of MODIS we felt it was better and simpler just to train and compare on the best version of MODIS available.

**RC3-14: Line 141. How are standardized anomalies calculated?**

*"Z-score" standardised anomalies are calculated as (x - m) / s where x is a monthly NDVI observation, m is the long-term mean NDVI for the given month, and s is the long-term standard deviation in NDVI for the given month. It is a standard way to track inter-annual variability. We would prefer not to include the formula in the manuscript as it is a very common approach, but we will include a reference to a published definition.*

**RC3-15: Line 146. More details are needed for the outperformance of GBM. For example, are all the models optimized in parameters?**

*Yes, all models were optimised and the GBM model outperformed them in terms of fitting accuracy and speed.  We did not include further information on discarded methods as they are not critical to the paper, but for those interested we have made the scripts for the GAM method available here along with the other code). We only wished to include this sentence so the reader would be aware that we had done our due diligence by testing on a few different methods before selecting gradient-boosting.*

**RC3-16: Line 152-153. "…in the heavily forested regions where there was little to no agreement between NDVIMCD43A4 and NDVIAVHRR…". How was pixel quality considered in calculating agreement?**

*All AVHRR and MODIS products were masked with their corresponding pixel-quality layers during loading and temporal compositing so only good-quality observations are retained. Then, when comparing one product to another, data gaps between all products were matched, as detailed in the comment above.*

**RC3-17: Line 155. Why is longitude not included? Give more details on NDVIMCD43A4 summary percentiles.**

*We had two reasons for this. Firstly, it proved not to be a particularly useful feature in the predictions (as evidence by low feature importance). Secondly, in an early version of the product, including both latitude and longitude introduced some artefacts into the predicted values. Ultimately, longitude is prone to overfitting. We did not feel this was important information to share with readers (since we also tried other features that did not make it into the final model configurations).*

*MODIS summary percentiles were calculated per pixel over the 2000-2022 period. So over the 22 year time-span, we extracted the 5th, 50th (median), and 95th percentile values. For the training and predictions, these values are simply replicated at each time step, so they are effectively static layers (i.e., not varying through time). We will include an extra note in an updated version of the manuscript describing the time-range over which the percentiles are calculated.*

**RC3-18: Line 178. Please list the hyperparameter values used.**

*Thank you for the suggestion. We will include a table in the appendix with the hyperparameters used for fitting the harmonisation models and the synthetic NDVI models.*

**RC3-19: Line 180. In addition to absolute error, a measure of error that reflects the relative error is also needed. Such a measure is particularly important for dense vegetation.**

*We will add RMSE to the statistics in the scatter plots of Figure 4 and Figure A5. However, arguably this information is already in the manuscript as the per-pixel coefficient-of-variation plots in figure 5b and 5d show relative error by dividing RMSE by the long-term mean NDVI.*

**RC3-20: Line 185. How are the long gaps spatially and temporally distributed, particularly for dense vegetation?**

*We argue that Figure A1 does a reasonable job of showing how data gaps are distributed spatially. We could potentially devise an additional figure that shows how gaps are temporally distributed, but we are not sure how much value this will add to the manuscript and are somewhat wary of adding too many figures. The time series in Figure 6 does show data gaps for two forested regions.*

**RC3-21: Line 191-192. What do you mean by methods in the bracket?**

*We are not entirely sure what the reviewer is referring to here, but the methods in the brackets refer to common spatial interpolation methods that can be used to fill gaps in data by extending values from nearby data points. There are quite a few methods available for this, some of which we listed in the brackets.*

**RC3-22: Line 198-199. Linear temporal interpolation may under or over-estimate values for seasonal peaks or valleys or other abrupt signals.**

*We completely agree with the reviewer, which is why we limited linear temporal interpolation to a single time step. This limits over or under estimation of temporal dynamics to a minimum.*

**RC3-23: Line 206-207. Why is not WCF used as a feature in data harmonization but in synthesis?**

*We did not wish to use WCF in the harmonisation as that product is partly built with the use of annual Landsat composites. Since we wanted to use Landsat as a validation of inter-annual variability in the pre-MODIS era, we felt using WCF might bias the results.*

**RC3-24: Line 219. Will there be any issue related to the calculation of phenology when up-sampling from monthly to two-week intervals?**

*The day-of-year values for POS are sensitive to temporal resolution since, at the monthly time scale, the POS value is really 'month-of-year' rather than 'day-of-year'. Upsampling is thus required to increase the temporal resolution and resolve any inter-month shifts in phenology. Upsampling of this magnitude is fairly common practice in*

remote sensing land-surface-phenology studies, and given this applications section is not the major focus of the study, we would suggest that a sensitivity analysis may overburden the manuscript. Note that in section 2.4 we highlight that DOY values are only an approximation.

**RC3-25: Line 238-239. How was the comparison made if there are data gaps brought by, for example, clouds? What if there are insufficient valid data between 2000 and 2013 for the calculation of CV and R?**

*This point was addressed in RC3-11(data gaps are matched between all datasets before comparisons are made). Even in relatively low data volume areas, there are still enough good quality observations to perform statistical calculations. For example: 14 years \* 12-months/per year \* 0.35 (a low fraction of data) = 59 good observations.  The exact procedures we used can be found in this jupyter notebook, in the github repository referred to in the manuscript.*

**RC3-26: Line 242. R2 (in the text) or R (in the figure)?**

*Thanks for pointing out this mistake, we will correct it.*

**RC3-27: Line 256-257. Present the length of the growing season please.**

*We are not sure how informative this would be but can add it.*

**RC3-28: Line 279-280. Solid evidence is required.**

*We appreciate the comment. Please refer to the discussion and figures in RC3-3 and RC2-2.*

**RC3-29: Figure 5. It would be interesting to see a similar residual NDVI map for NDVIpku.**

*There is a low residual signal between MODIS and GIMMS-PKU-consolidated, as one would expect based on the agreement maps in figure 2d and 2h. We argue that including this in the manuscript would not add much value - we did not contest that GIMMS-PKU-consolidated agrees well with MODIS. Figure 5 is intended to show the results of our calibration and harmonisation, not GIMMS-PKU's.*

**RC3-30: Figure 6. Notice that the increased trend of NDVI before 2000 in AVHRR-CDR disappears in AusE-clim.**

*The trend in CDR is almost certainly an artificial artefact of step changes between sensor transitions and poor calibration over these regions. Below we replot the same figures but including GIMMS3g which has had sensor transitions ameliorated and the trend slope is much less than for CDR in either of the two regions plotted. Similarly, MODIS over the 22-year period does not show trends like those of CDR, yet the likely drivers of greening ($CO_2$ and warming) all continue to increase from 2000-2022.  As these plots are intended to show 'before-and-after' calibration results, we would prefer not to muddle them by including additional time series. However, we could include one of these time series in the appendix and make a note in the results sections that the artificial trend in CDR is removed by the GBM calibration, at the editor's discretion.*

[Figure]

**RC3-31:** **Figure 7. Focus needs to be placed on vegetated, particularly densely vegetated areas. Also, in Figure 7e, is the red dot line calculated without any observation data?**

*Yes, the synthetic NDVI data plotted in Figure 7e is created using only climate data, MODIS summary percentiles, and annual WCF - averaged over Australia the synthetic data does a great job of replicating observations.*

*In an updated version of the manuscript we can update Figure 7 to include a second time-series showing the synthetic data over a densely vegetated region, similar to figure 6.*

**RC3-32:** **Line 370. What do you mean by 'gaps in the NDVIPKU-consolidated dataset'? Non-data or data with poor quality?**

*We mean some pixels have no data because the QC layers that come with GIMMS-PKU labelled these pixels as poor-quality observations. Specifically, for GIMMS-PKU we kept only those pixels labelled as 'good-quality AVHRR'. And for GIMMS-PKU-consolidated we kept only those pixels labelled as 'good-quality AVHRR' and 'good-quality MODIS' and where the harmonisation was run by the random-forest model. We will update the data section to include more information on the QC masking procedures.*

**RC3-34:** **Figure 8. Note that NDVIpku is generated from a different MODIS NDVI product. A comparison between MODIS NDVI products may be beneficial.**

*Noted, but we do not wish to overwhelm readers by adding further figures and discussion of inter-comparing versions of MODIS. We argue this would not add much value to the manuscript. As stated previously, we agree that GIMMS-PKU-consolidated agrees well with MODIS. We will also update Figure 8 to include the anomaly time-series shown in our response to RC3-3 and RC2-2. As anomalies remove the mean value, the agreement between GIMMS-PKU and MODIS will narrow, and the focus instead will be on differences in inter-annual variability.*

*References*

*Byrne, G., Broomhall, M., Walsh, A. J., Thankappan, M., Hay, E., Li, F., ... & Denham, R. (2024). Validating Digital Earth Australia NBART for the Landsat 9 Underfly of Landsat 8. Remote Sensing, 16(7), 1233.*

---

## Author Response (AR2)

**Response to Reviewers 30/7/24**

**Editor**

**Please consider all the reviewers' comments and address their concerns as much as possible.**

**Additional private note (visible to authors and reviewers only):**
**Although two reviewers have suggested minor revisions for the manuscript, the third reviewer has recommended rejection. Please ensure that you thoroughly address the concerns raised by the third reviewer.**

*Please find below our responses to each reviewer in italics. As per our previous responses, we have labelled each significant reviewer comment with the code [reviewer]-[comment number], e.g. RC1-1.*

*We have provided detailed and thorough responses to reviewer three. In our opinion, the reviewer has not provided strong evidence for their claims, and we hope you find the reasoning and evidence in our rebuttals persuasive.*

*Note also that we have made a minor update to the datasets on the Zenodo repository. These updates include only very minor changes and do not impact the results of the manuscript in any way. The changes are as follows:*
1. *All datasets now have their NDVI values clipped to the range 0-1. This step was included when analysing datasets in the manuscript, but the originally uploaded Zenodo datasets hadn't been clipped.*
2. *The AusENDVI-clim dataset is now gap-filled, and includes a QC layer.*
3. *The merged AusENDVI-noclim_MCD43A4_1982_2022 dataset was removed to simplify the number of datasets included in the repository. Users who want to join the noclim and MCD43A4 datasets can do so by clipping out MCD43A4 from the AusENDVI-clim_MCD43A4_gapfilled_1982_2022 dataset.*
4. *The accompanying Jupyter Notebook 'readme' has been updated.*

**RC1**

**Authors did a good job responding to my concerns. I reread the paper in detail and it feels much more convincing and thorough. The following comments below are minor and, after their consideration, the manuscript should be ready for publication.**

*We thank the reviewer again for their comments in the first round of revisions and we are pleased to have addressed their initial concerns.*

**RC1-1 Line 31: "shows excellent agreement with observations", but what observations?**

*The synthetic NDVI dataset is compared with both MODIS and the recalibrated AVHRR datasets. We will update the text to be more specific here.*

**RC1-2 Line 33: The term "seamlessly joined" is somewhat uncertain and reminds me of a recent paper (https://doi.org/10.1029/2023EF004119). In this study, authors used a simple statistical approach to identify abrupt shift points in merged NDVI/LAI products (Fig. S7), suggesting more abrupt shift points in merged products compared to non-merged ones. Therefore, I recommend including this analysis as evidence supporting the "seamlessly joined" nature of AusENDVI.**

*We agree with the reviewer that 'seamlessly joined' is a vague phrase and we have removed the term 'seamlessly' from the text. The merging of AusENDVI with MODIS MCD43A4 is of course not perfectly seamless, especially in those regions where we identified lower correlations and higher error in the calibration (e.g., irrigated cropping regions, snow impacted high elevation regions). Our claim is really that AusENDVI is better calibrated to MODIS in terms of mean NDVI, interannual variability, and seasonal variability, than currently existing products and this warrants the joining of the datasets to create a >40 year record of vegetation condition.*

*We are reluctant to include a breakpoint analysis in the paper for two main reasons. Firstly, it is uncertain if the statistical approach shown in the linked paper could distinguish between breakpoints due to merging issues, or due to abrupt shifts in the climate, as breakpoint analysis is used for both purposes (de Jong et al. 2011; Tian et al. 2015). For example, at the transition between AVHRR and MODIS in the year 2000, Australia underwent a* very strong La Niña event *(with anomalously high rainfall over much of the continent) that resulted in widespread greening across Australia, this anomalous greening was immediately followed by a decade long drought (the "Millennium" drought) that led to widespread reduction in vegetation productivity (Van Djik et al. 2013). Secondly, we are reluctant to include more analysis and figures into an already long manuscript (currently 23 figures and tables). On Zenodo, we provide both the merged AusENDVI and MCD43A4 product, along with the unmerged AusENDVI datasets that span 1982-2013 (effectively re-calibrated Climate Data Record v5 AVHRR datasets for Australia). If a user is concerned about an artificial break at the join between AusENDVI and MODIS, then they can rely upon the unmerged products.*

*It's worth noting also that products such as GIMMS-PKU, which ostensibly have not undergone merging, are not immune to sudden and dramatic changes between generations of sensors. The figure in our response to RC3-3 (in this round of revisions) shows a dramatic 'break point' after the year 2000 in the long-term trend of inter-annual variability.*

**RC1-3 Line 70: the inconsistent findings can also include the interannual variability of vegetation greenness (https://doi.org/10.1029/2023EF004119).**

*Thank you for the link to this paper, we will update the introduction to include reference to this recent paper.*

**RC1-4 Line 105: add citations regarding resampling techniques.**

*We would be happy to include a citation here but we are not sure who to reference for nearest-neighbour and average resampling techniques. Moreover, these are very well known geospatial processing techniques available in most, if not all, image processing software and packages.*

**RC1-5 Figure 1: please use the standard flowchart symbols and notation.**

*We have updated Figure 1 in the manuscript with standard flowchart symbols.*

**RC1-6 Line 261: extra "." at the end of sub-title 3.1**

*We have removed this in the updated manuscript.*

**RC1-7 Lastly, I suggest that the authors thoroughly review the paper and consider shortening certain sections (e.g., abstract and research objectives) to maintain clarity and conciseness.**

*We have edited the abstract and research objective for conciseness.*

*References*
*De Jong, R., Verbesselt, J., Schaepman, M. E., & De Bruin, S. (2012). Trend changes in global greening and browning: contribution of short-term trends to longer-term change. Global Change Biology, 18(2), 642-655.*

*Tian, F., Fensholt, R., Verbesselt, J., Grogan, K., Horion, S., & Wang, Y. (2015). Evaluating temporal consistency of long-term global NDVI datasets for trend analysis. Remote Sensing of Environment, 163, 326-340.*

*Van Dijk, A. I., Beck, H. E., Crosbie, R. S., De Jeu, R. A., Liu, Y. Y., Podger, G. M., ... & Viney, N. R. (2013). The Millennium Drought in southeast Australia (2001–2009): Natural and human causes and implications for water resources, ecosystems, economy, and society. Water Resources Research, 49(2), 1040-1057.*

**RC2**

**Thanks to the author's efforts, which solved all my problems, I suggest that ESSD could accept the article. I still have some minor issues:**

*We thank the reviewer again for their comments in the first round of revisions and we are pleased to have addressed their initial concerns.*

**RC2-1 Sources of global vegetation change include direct effects due to CO2 fertilization effects and indirect impacts due to climate change because of radiative effects. I am curious if changes in CO2 levels and current changes in NDVIs are an**

**identical or similar trend and I think this is worth exploring. The authors discuss the correlation of climate, especially precipitation, with NDVIs, and it might be good to include the correlation of CO2 changes and NDVI trends in the discussion section. I have doubts about whether NDVI can accurately reflect these kinds of complex changes, and I hope the authors can discuss this point.**

*We agree with the reviewer that the relationship between increasing atmospheric $CO_2$ and greening of vegetation is an important consideration. The reliance on satellite remote sensing of vegetation to attribute greening trends to $CO_2$ fertilisation and/or climate change has at times led to misattribution of trends where fusing of datasets has been poor (Wang et al. 2020). Likewise, remotely sensed vegetation condition datasets have been used to argue for fundamental changes to ecosystems in response to climate change (Donohue et al. 2013; Poulter et al. 2014). Yet, it's possible these changes have been overstated owing to spurious trends in inter-annual variability (Tian and Luo 2024). We argue that AusENDVI allows for a more robust addressing of these questions. As we wish to investigate some of the questions in subsequent work, we do not wish to pre-empt future work by including further analysis and discussion here, especially given the already considerable length of the manuscript.*

*References*
*Wang, S., Zhang, Y., Ju, W., Chen, J. M., Ciais, P., Cescatti, A., ... & Peñuelas, J. (2020). Recent global decline of CO2 fertilization effects on vegetation photosynthesis. Science, 370(6522), 1295-1300.*

*Tian, J., & Luo, X. (2024). Conflicting changes of vegetation greenness interannual variability on half of the Global vegetated surface. Earth's Future, 12(5), e2023EF004119.*

**RC3**

**RC3-1 In previous comments, I expressed my deep concerns and hoped the authors would greatly improve their dataset and manuscript with the points that have been elaborately listed. However, it's disappointing that far-less-than-expected improvements have been made in the new version. This time I'll point out major ones.**

*In their previous comments the reviewer provided a detailed evaluation of the manuscript, providing 33 comments. We provided detailed responses to every comment: for 13 of these comments we provided a general response, and for 20 of these comments we responded with changes to either the manuscript text, or through the inclusion or updating of figures and tables (we added four new figures and one new table, and updated three other figures). We believe this constitutes a significant enhancement to the manuscript and we are grateful to the reviewer for making it possible. We also note that two other reviewers considered our revisions as substantial and are now satisfied with the manuscript.*

**RC3-2 A new (or enhanced) dataset must show its significance or distinctiveness, for example, a new type of earth variable, wider spatiotemporal coverage, higher spatiotemporal resolution, or higher data accuracy in either absolute value or spatiotemporal patterns. This is what the authors failed to demonstrate in their**

**AusENDVI. The author claimed a 'comprehensive' evaluation of existing NDVI datasets in the abstract, but the evaluation is not.**

*Vegetation within semi-arid and arid ecosystems, of which the Australian continent is dominated, play a critical role in controlling the IAV of the global carbon dioxide growth rate (Donohue et al. 2013, Ma et al. 2016, Poulter et al. 2014, Zhu et al. 2016). Remotely sensed vegetation condition datasets have also been used to argue for fundamental changes to ecosystems in response to climate change (Donohue et al. 2013, Poulter et al. 2014). Thus, NDVI is an exceptionally important metric, too important to be left to just a few consolidated datasets, especially when both we and many others have shown that the commonly used global datasets contain substantial limitations. Having independent algorithmic approaches to creating a harmonised data set is important, and is a significant contribution. We have concentrated on making a dataset that works well for the Australian context where vegetation conditions are characterised by high interannual variability (discussed further below in RC3-3). Furthermore, we argue the manuscript is valuable not just in terms of the datasets created and their downstream applications, but also because the open-source and reproducible methods provide an iterative advancement. Though we focussed our study on the regional rather than global scale, we welcome others who may see merit in our methods and seek to apply them to other regions or the globe, either in terms of the algorithms used for harmonisation, the feature datasets used, or the gap-filling procedures.*

*On the evaluation of existing NDVI datasets: our manuscript utilises five global NDVI products. We evaluate the two most commonly cited global AVHRR NDVI datasets in the literature (GIMMS3g, CDR/LTDR), and the most recent global AVHRR dataset (GIMMS-PKU). The manuscript includes a total of seven figures (including the appendix) where various NDVI products are inter-compared. Whether or not this counts as 'comprehensive' is subjective, but we argue the assessment of three popular global AVHRR NDVI datasets provides substantial value to a reader who may be interested in using a long-term remotely sensed NDVI product to chart changes to vegetation conditions in Australia.*

**RC3-3 From the comparison between AusENDVI and PKU-consolidated, I noticed a comparable spatial resolution (0.05 vs 1/12=0.083), interannual variability (Figure 9a, despite the authors arguing a significant improvement), and phenology (Figure 3k and Figure 7c; considering the primal difference between MOD13C1 and MCD43A4).**

*On spatial resolution, we argue that a ~25 km$^2$ pixel is a substantial improvement over a ~64 km$^2$ pixel. The 0.05 degree resolution also brings the NDVI dataset in line with high quality regional meteorological datasets, of the kind used to consider relationships between the climate and vegetation (Jones et al. 2009).*

*The consistency of interannual variability (IAV) in AusENDVI is substantially improved over GIMMS-PKU. We argue that we have shown this clearly in Figure 9 where PKU does not track the IAV of Landsat, and PKU's sensitivity to rainfall more than doubles coinciding with the transition from the AVHRR era (1982-2000) to the MODIS era (2000-2022). To demonstrate this further, Figure 1 (below) maps the trends in IAV as indicated by the coefficient of variation (CV), following the methods described in Tian and Luo (2024). PKU shows a distinct lack of IAV in the pre-MODIS era (and no trend in IAV), then the IAV more than doubles (~0.05 to 0.115) from the early 2000's onwards (the figure shows 10-year*

*rolling mean CV where each time-stamp is centred in the middle of the 10-yr period). This dramatic jump in IAV coinciding with the switch to better quality AVHRR sensors from the early-to-mid 2000's onwards is highly unlikely to be natural, and indicates a substantial limitation in the PKU product (over Australia, it may be fine elsewhere - we haven't assessed it globally). Contrast this with AusENDVI-clim which shows a gradually increasing IAV through the 90's and into the 2000's that is far more realistic considering the likely impacts of steadily increasing atmospheric $CO_2$ and its role in increasing the water use efficiency of plants (Donohue et al 2009, Rifai et al. 2022).*

[Figure]

*Figure 1: Temporal changes in the rolling 10-year average coefficient of variation (standard deviation / mean) indicating trends in inter-annual variability. CV was calculated per-pixel and then averaged over the continent.*

*On phenology, we agree that for the example regions shown in the manuscript, GIMMS-PKU and AusENDVI perform similarly, and we do not dispute this point in the manuscript.*

**RC3-4 However, (a) AusENDVI was not compared to individual reference datasets in terms of absolute value.**

*AusENDVI was cross-validated with MODIS MCD43A4 in terms of absolute NDVI values (see Figure 5; both MAE and RSME statistics are reported). This is the same approach as was taken for validating the GIMMS-PKU dataset, though they cross-validated against Landsat because that was their target dataset (Li et al. 2023). There are no ground-truth NDVI values that we could rely upon as reference datasets.*

**RC3-5 Landsat data were used in this study, but not for absolute value evaluation. Discrepancies between Landsat-5 and Landsat-7 were not cross-calibrated (as I mentioned but the authors did not try to solve). In the response, the authors argued in the Figure of RC3-7 (no Figure number provided) that the average line sits between the lines. This is obviously, not true. The average line basically overlaps Landsat-7, which means Landsat NDVI anomalies in the Landsat-7 era are systematically higher than the Landsat-5 era and the Landsat baseline itself may not be accurate.**

*There is no reason why absolute Landsat NDVI values should agree with AusENDVI values as the spectral sampling between the sensors is different, as the reviewers argued themselves in their earlier response (RC3-1 in the first review). We instead rely on Landsat as a validation of inter-annual variability in the pre-MODIS era as differences in absolute*

*NDVI values should not impact the annual temporal variability of the series. We showed that Landsat anomalies agree well with MODIS anomalies from 2000-2012 (Figure A2; included at the request of the reviewer), and that AusENDVI agrees well with Landsat anomalies from 1988-2012 (Figure 9a), indicating that AusENDVI can reliably reproduce climate driven variability in both in the pre-MODIS and post MODIS eras. This finding highlights the product's usefulness as a long-term indicator of the vegetation response to climate (change) - a key use case for long-term remotely sensed NDVI datasets.*

*The small differences in mean NDVI between Landsat 5 and Landsat 7 are immaterial to the use case here, which we argued convincingly in the first review (RC3-7). To reiterate, firstly, the absolute difference in NDVI values between the Landsat 5 and Landsat 7 time series shown in the earlier response (RC3-7; the figure is reproduced below for convenience, Figure 2) is equal to 0.004 NDVI, a negligible difference. As we are using Landsat to evaluate temporal variability, the range in NDVI values is arguably more important than the mean, and the range difference between Landsat 5 and Landsat 7 in Figure 2 are also negligible, equal 0.002 NDVI. Secondly, within each month we combine all Landsat 5 and 7 observations together using a median composite so the small difference between them is further ameliorated. Thirdly, we anomalize the data by subtracting the long-term mean so absolute differences in Landsat NDVI are irrelevant. We argue the reviewer has not made a convincing case as to why Landsat should not be suitable for assessing the IAV of AusENDVI in the pre-MODIS era.*

[Figure]

*Figure 2: Landsat 5 and Landsat 7 over Australia for 2004, the first year of full overlap in the series. The Landsat data product is Collection 3 Landsat NBAR provided by Digital Earth Australia.*

**RC3-6 The time interval for PKU-consolidated is half a month but for AusENDVI is a month. The temporal resolution is critical for tracking the quick change of vegetation under current climate change. Despite the authors arguing that a month is enough, many others have pointed out that half a month is not enough. Noting PKU-consolidated is a global dataset and AusENDVI is a regional one, the two flaws mentioned above are unacceptable.**

*In satellite images, high frequency variation is very often due to noise, and one means of ameliorating this noise is to calculate monthly composites to minimise spurious changes due*

to clouds etc. Monthly resolution datasets are commonly used for long term studies of climate and vegetation (Gonsamo et al. 2020; Rifai et al. 2022) and we see no reason why the monthly resolution datasets of AusENDVI cannot be relied upon for this purpose.

*The reviewer has not provided any references or evidence to back up their claim that "...many others pointed out that half a month is not enough" for land-surface phenology (LSP) studies. LSP studies commonly start with fortnightly or monthly data and temporally upscale the datasets using curve fitting or interpolation techniques (Zeng et al. 2020). Users of AusEDNVI can do the same.*

*The reviewer did not provide references or evidence for the statement that "...temporal resolution is critical for tracking the quick change of vegetation under climate changes". Changes to vegetation dynamics from climate change are not 'quick' but rather take decades as plants respond to long term trends in temperature, $CO_2$, and rainfall patterns (Donohue et al. 2013; Garonna et al. 2015; Zhu et al. 2016; Winkler et al. 2021). Thus the need for reliable, very long term datasets to extract subtle trends in vegetation in response to climate change amidst the 'noise' of interannual and interdecadal variability.*

**RC3-7 The significance of AusENDVI should have been a more accurate characterization of rainfall in the specific country of Australia. I have raised it as a major comment, but the authors simply provided a correlation map with low correlation values in many regions. The authors claim strong correlations but note that the values could be lower than 0.3. No significance values and further in-depth analysis were provided.**

*We are surprised that the reviewer remains unconvinced that Australian ecosystems are predominantly water-limited given approximately seventy percent of Australia's land mass is classified as arid and semi-arid (see the Köppen climate classifications here).  In the manuscript we provide three references that outline Australia's water-limited nature (Peters et al., 2021; Poulter et al., 2014; Broich et al., 2014), and the per-pixel correlation map shows most areas over Australia NDVI correlates with rainfall between 0.5 - 0.9 R (Figure 4c, this was included at the request of the reviewer).  Of course, not every location in Australia is water-limited (e.g. tropical rainforests in Queensland, wet temperate forests in western Tasmania), but this does not negate the relationship between IAV in rainfall and IAV in vegetation condition over Australia when aggregated to the continental scale.*

**RC3-8 Moreover, when I re-evaluated the whole manuscript and tried to find (potentially) significant improvements raised not by me but by other reviewers, I figured out a newly introduced flaw. In Figure 9d, the authors tried to prove AusENDVI better follows variations in precipitation, by building a relationship between AusENDVI-clim and 12-month rolling rainfall. However, rainfall has been a contributor to the generation of AusENDVI-clim (section 2.3).**

*In Figure 9 we are not trying to show that AusENDVI has a higher correlation with rainfall ("better follows variations in rainfall"), but rather that AusENDVI's sensitivity to rainfall (the linear slope coefficient, i.e. how much NDVI responds per unit rainfall) is more consistent between the AVHRR and MODIS eras. This is in contrast to GIMMS-PKU which shows a more than doubling of sensitivity. This finding is further reinforced by Figure 1 in our*

*response above (RC3-3) where GIMMS-PKU shows a doubling of IAV (as shown through trends in the coefficient of variation) between the 1982-2000 and 2000-2022 periods. The inclusion of climate variables in the calibration is likely one reason why AusENDVI displays more consistent IAV throughout the time series, but we argue this is a positive development. Note also that the most important variable for the calibration and harmonisation models, by far, are the AVHRR NDVI values themselves, not rainfall, as shown through the SHAP feature importance plots in Figure A4 in the manuscript.*

**References**

*Jones, D. A., Wang, W., & Fawcett, R. (2009). High-quality spatial climate data-sets for Australia. Australian Meteorological and Oceanographic Journal, 58(4), 233.*

*Tian, J., & Luo, X. (2024). Conflicting changes of vegetation greenness interannual variability on half of the Global vegetated surface. Earth's Future, 12(5), e2023EF004119.*

*Donohue, R. J., Roderick, M. L., McVicar, T. R., & Farquhar, G. D. (2013). Impact of CO2 fertilization on maximum foliage cover across the globe's warm, arid environments. Geophysical Research Letters, 40(12), 3031-3035.*

*Garonna, I., de Jong, R., & Schaepman, M. E. (2016). Variability and evolution of global land surface phenology over the past three decades (1982–2012). Global change biology, 22(4), 1456-1468.*

*Winkler, A. J., Myneni, R. B., Hannart, A., Sitch, S., Haverd, V., Lombardozzi, D., ... & Brovkin, V. (2021). Slowdown of the greening trend in natural vegetation with further rise in atmospheric CO 2. Biogeosciences, 18(17), 4985-5010.*

*Gonsamo, A., Ciais, P., Miralles, D. G., Sitch, S., Dorigo, W., Lombardozzi, D., ... & Cescatti, A. (2021). Greening drylands despite warming consistent with carbon dioxide fertilization effect. Global Change Biology, 27(14), 3336-3349.*

*Rifai, S. W., De Kauwe, M. G., Ukkola, A. M., Cernusak, L. A., Meir, P., Medlyn, B. E., & Pitman, A. J. (2021). Thirty-eight years of CO 2 fertilization have outpaced growing aridity to drive greening of Australian woody ecosystems. Biogeosciences Discussions, 2021, 1-41.*

*Peters, J. M., López, R., Nolf, M., Hutley, L. B., Wardlaw, T., Cernusak, L. A., & Choat, B. (2021). Living on the edge: A continental-scale assessment of forest vulnerability to drought. Global Change Biology, 27(15), 3620-3641.*

*Poulter, B., Frank, D., Ciais, P., Myneni, R. B., Andela, N., Bi, J., ... & van der Werf, G. R. (2014). Contribution of semi-arid ecosystems to interannual variability of the global carbon cycle. Nature, 509(7502), 600-603.*

*Broich, M., Huete, A., Tulbure, M. G., Ma, X., Xin, Q., Paget, M., ... & Held, A. (2014). Land surface phenological response to decadal climate variability across Australia using satellite remote sensing. Biogeosciences, 11(18), 5181-5198.*

*Zeng, L., Wardlow, B. D., Xiang, D., Hu, S., & Li, D. (2020). A review of vegetation phenological metrics extraction using time-series, multispectral satellite data. Remote Sensing of Environment, 237, 111511.*

*Li, M., Cao, S., Zhu, Z., Wang, Z., Myneni, R. B., & Piao, S. (2023). Spatiotemporally consistent global dataset of the GIMMS Normalized Difference Vegetation Index (PKU GIMMS NDVI) from 1982 to 2022. Earth System Science Data, 15(9), 4181-4203.*

*Ma, X., Huete, A., Yu, Q., Coupe, N. R., Davies, K., Broich, M., ... & Eamus, D. (2013). Spatial patterns and temporal dynamics in savanna vegetation phenology across the North Australian Tropical Transect. Remote sensing of Environment, 139, 97-115.*

*Zhu, Z., Piao, S., Myneni, R. B., Huang, M., Zeng, Z., Canadell, J. G., ... & Zeng, N. (2016). Greening of the Earth and its drivers. Nature climate change, 6(8), 791-795.*